# Human-like Few-Shot Learning via Bayesian Reasoning over Natural Language

**Kevin Ellis**
Cornell University
`kellis@cornell.edu`

## Abstract

A core tension in models of concept learning is that the model must carefully balance the tractability of inference against the expressivity of the hypothesis class. Humans, however, can efficiently learn a broad range of concepts. We introduce a model of inductive learning that seeks to be human-like in that sense. It implements a Bayesian reasoning process where a language model first proposes candidate hypotheses expressed in natural language, which are then re-weighed by a prior and a likelihood. By estimating the prior from human data, we can predict human judgments on learning problems involving numbers and sets, spanning concepts that are generative, discriminative, propositional, and higher-order.

## 1 Introduction

Human learning is rapid and broad. Consider a child learning a new routine like 'high-five': given just 1 or 2 examples, they can learn that move, and even generalize to variations like low-fives. Or consider learning the basics of a game like Pacman, or an AI researcher learning a new technique like 'few-shot prompting', or a mathematician extrapolating '1, 4, 16, 64, ...'. In each of these cases, the relevant routine, rules, or concept can be learned from little data and only seconds to minutes of experience. Furthermore, the space of possible concepts is effectively infinite, because concepts can compose to yield more complex constructs like 'high-five followed by a fist bump'. Thus a key computational challenge is to understand how an intelligent system can acquire a wide range of new concepts, given modest data, and granted a modest computational budget. Building AI systems that can efficiently master many concepts is also practically valuable, because data-efficiency and broad generalization remain some of the most salient gaps between human and machine intelligence [1].

Fundamentally, we are concerned with the problem of *induction*: Inferring a generalizable pattern, rule, trend, or law from specific examples. A classic approach to induction is to start with a hypothesis space of possible concepts, and then probabilistically infer which hypothesis most likely generated the observed data using Bayes' Rule. This Bayesian paradigm has proved widely applicable across both cognitive science and artificial intelligence [2, 3, 4, 5, 6, 7, 8, 9, 10, 11, 12, 13, 14].

On its own, however, the Bayesian paradigm leaves important questions unanswered. Acquiring a broad range of possible concepts demands an expressive hypothesis space, but inference over a rich space of concepts comes at steep computational cost. For increased expressivity, many Bayesian models design hypothesis spaces that resemble a programming language [15, 16, 17, 18, 19, 20]. Posterior inference then corresponds to constructing high-probability programs. Each of these program-learning models requires a custom domain-specific programming language. Despite confining themselves to a domain-specific language, these models still require specialized inference machinery, such as heuristically-designed search moves [10], or exorbitant compute budgets [21].

Our goal is to build a model of humanlike concept learning that makes progress toward resolving the tension between intractable inference and expressive hypothesis classes. We propose a new model

37th Conference on Neural Information Processing Systems (NeurIPS 2023).

that expresses its concepts in natural language–even when the learning problem does not involve natural language. We do this for two reasons. First, language is an effective representation for many human concepts. It is compositional, richly expressive, and regularizes the learner toward natural generalizations. Second, we find we can efficiently infer natural language concepts using modern large language models [22, 23] based on the transformer architecture [24].

Like any Bayesian model, we will first define a prior over concepts, which in our case exerts a top-down pressure for naturally-structured language. Our model also has a bottom-up mechanism for efficiently inferring possible hypotheses, analogous to a recognition network [25]. The interaction of these top-down and bottom-up models is surprisingly effective as a model of humanlike learning: Given around 50 samples from the recognition network, our model can account for human patterns of generalization for concepts that are generative or discriminative, and propositional or first-order. We show the model can also capture fine-grained structure in human judgements: both subtle gradations of uncertainty, and also the dynamics of learning starting from a few examples and going to dozens of examples. Finally, a key reason why humans can learn so efficiently is because they have a good inductive bias or prior [26, 27]. Our model can fine-tune its prior to human judgements, effectively extracting a human-like prior from behavioral data. We find this gives a more faithful model of human generalization, and also improves average accuracy on concept-learning tasks.

We focus on abstract symbolic concepts, such as 'prime numbers less than 30', which we think are well-suited to natural language. We do not consider concepts grounded in perception and actuation, such as 'dog' or 'chewing', and hence do not provide a single unified account of inductive learning.

However, for these abstract concepts, we provide a *rational process model* [28]. Following the Marr levels of analysis [29], this means we propose algorithmic mechanisms for concept learning that rationally approximate optimal inference, subject to bounds on computation (sampling). This contrasts with *computational-level models*, which characterize the goal of the learner, without committing to a theory of how the learner mechanistically accomplishes that goal [29]. Most Bayesian concept learning models operate at the computational level, avoiding issues of intractability [3, 15] (cf. [30]).

We contribute (1) a model of symbolic concept learning that supports efficient inference over a flexible hypothesis class; (2) an evaluation on human data from two different concept learning experiments; and (3) a simple recipe for extracting a humanlike prior over concepts, given raw behavioral data.

## 2    Background and Related Work

**Bayesian Program Learning**  (BPL: [10]) models concept learning as Bayesian inference over latent symbolic programs. BPL models first define a domain-specific programming language spanning a space of possible concepts, and then infer a posterior over concepts $C$ given training data $D$ via Bayes' Rule: $p(C|D) \propto p(D|C)p(C)$. Although the set of possible concepts remains hardcoded, the prior $p(C)$ can be learned through hierarchical Bayesian methods. Given parameters $\theta$ indexing possible priors, and a collection of datasets $\mathcal{D}$, the prior can be learned by approximately solving [21]:

$$\theta^* = \arg\max_\theta p(\theta|\mathcal{D}) = \arg\max_\theta p(\theta) \prod_{D \in \mathcal{D}} \sum_C p(D|C)p_\theta(C)$$

Bayesian models can learn from few examples, because the prior regularizes them toward reasonable generalizations. They also produce nuanced uncertainty estimates, because they represent the posterior. Program learners can also acquire any concept expressible in their domain-specific language, but this language must be appropriately circumscribed for inference to remain tractable. Systems such as DreamCoder [21] partly address this concern by growing the language and training neural networks to aid inference, but even then, general-purpose programming languages remain out of scope. We next turn to latent language, which considers the broad hypothesis space of all natural language.

**Latent Language.**    Using language as an internal representation for nonlinguistic tasks was introduced by the Latent Language framework [31, 32]. These systems perform few-shot learning by searching for the language which minimizes a loss on the training examples. Given training input-output examples $\{(x, y)\}$, the latent language approach infers the language $C^*$ minimizing

$$C^* = \arg\min_{C \in \Sigma^*} \sum_{(x,y)} \text{Loss}(y, f_\theta(x; C))$$

where $f_\theta$ is a neural network and $\Sigma^*$ is the set of all strings. Because there are infinitely many strings, another neural network samples a finite pool of candidate concepts. Relative to Bayesian learners,

latent language models use maximum likelihood estimation to infer a single concept, rather than construct a posterior distribution. Like our approach, latent language uses language as an intermediate representation, combined with a bottom-up concept proposal process. Our work adds additional Bayesian mechanics, and shows how to learn a prior on $C$ from human judgments. Learning the prior proves important for both modeling human judgments and achieving high task performance.

**Induction, Abduction, and Deduction.** We address inductive problems: inferring a general pattern from specific examples [33]. Abduction is a related process where the reasoner infers an explanation for a specific observation. Abductive reasoning using modern language models has received much recent attention [34, 35], which has solidified natural language as a promising candidate for representing abductive explanations. Deduction—logically inferring the truth of a proposition—has also have been similarly revisited in the context of modern language models [36, 37].

## 3 Model

We start with a basic Bayesian approach. A latent concept $C$ generates $K$ observed examples, notated $X_1, \ldots, X_K$, according to an IID process. We abbreviate $X_1, \ldots, X_K$ as $X_{1:K}$. For our model, $C$ is an utterance in natural language. The learner's posterior beliefs are given by Bayes's Rule,

$$p(C|X_{1:K}) \propto p(C) \prod_{1 \le k \le K} p(X_k|C) \tag{1}$$

The prior $p(C)$ comes from a neural model. The likelihood $p(X|C)$ is domain-specific because it depends on the structure of the examples. We assume the posterior in Eq. 1 is intractable.

We model tasks that do not involve externalized language, thus beliefs over $C$ play a fundamentally auxiliary role. We instead care about the probability that a test example $X_{\text{test}}$ belongs to the same concept as the training examples $X_{1:K}$. This posterior predictive quantity is

$$p(X_{\text{test}} \in C|X_{1:K}) = \sum_C p(C|X_{1:K}) \mathbb{1}\left[X_{\text{test}} \in C\right] \tag{2}$$

To make the above tractable, we introduce a proposal distribution $q(C|X_{1:K})$. We draw from $q$ a modest number of sampled concepts (tens to hundreds), writing those samples as $C^{(1)}, \ldots, C^{(S)}$. By only considering concepts proposed by $q$, the infinite sum over $C$ in Eq. 2 becomes a finite sum over $S$ samples. Provided those proposals account for much of the posterior mass, this is a good approximation. Conventionally, $q$ is used to construct an importance sampler [38]:

$$p(X_{\text{test}} \in C|X_{1:K}) = \mathop{\mathbb{E}}_{C \sim p(\cdot|X_{1:K})} \mathbb{1}\left[X_{\text{test}} \in C\right]$$

$$\approx \sum_{1 \le s \le S} w^{(s)} \mathbb{1}\left[X_{\text{test}} \in C^{(s)}\right], \text{ where } w^{(s)} = \frac{\tilde{w}^{(s)}}{\sum_{s'} \tilde{w}^{(s)}} \text{ and } \tilde{w}^{(s)} = \frac{p(C^{(s)})p(X_{1:K}|C^{(s)})}{q(C^{(s)}|X_{1:K})} \tag{3}$$

The above Monte Carlo estimate requires evaluating $q(C^{(s)}|X_{1:K})$. The most powerful proposal distributions at our disposal, such as GPT-4 [22], do not expose this functionality, so we heuristically approximate importance sampling by deduplicating the samples and weighing each by $p(C)p(X_{1:K}|C)$:

$$p(X_{\text{test}} \in C|X_{1:K}) \approx \sum_{C \in \{C^{(1)}, \ldots, C^{(S)}\}} w^{(C)} \mathbb{1}\left[X_{\text{test}} \in C\right], \text{ where }$$

$$w^{(C)} = \frac{\tilde{w}^{(C)}}{\sum_{C'} \tilde{w}^{(C')}} \text{ and } \tilde{w}^{(C)} = p(C)p(X_{1:K}|C) \mathbb{1}\left[C \in \{C^{(1)}, \ldots, C^{(S)}\}\right] \tag{4}$$

Ultimately, the distribution $p(C)$ should reflect the prior beliefs of human learners. To tune the prior to reflect human patterns of generalization, we assume access to a dataset of human judgments consisting of triples $(X_{1:K}, X_{\text{test}}, r)$, where $r$ is the ratio of humans who judged $X_{\text{test}}$ as belonging to the same concept as $X_{1:K}$. More generally, $r$ could be any average human rating in $[0, 1]$. If $\theta$ parametrizes the prior, then we match the prior to the human data by solving

$$\arg\max_{\theta} \sum_{(X_{1:K}, X_{\text{test}}, r)} r \log p_\theta(X_{\text{test}} \in C|X_{1:K}) + (1 - r) \log\left(1 - p_\theta(X_{\text{test}} \in C|X_{1:K})\right) \tag{5}$$

where $p_\theta(X_{\text{test}} \in C|X_{1:K}) = \sum_C \mathbb{1}\left[X_{\text{test}} \in C\right] p_\theta(C|X_{1:K})$, approximated via Eq. 3-4

As a process model, the approach claims that a bottom-up generator proposes a bounded number of candidate concepts, which are re-weighed by re-examining each datapoint, while being biased by a prior. This can be seen as a dual-process model [39], as well as a model of bounded rationality [40].

**Comparing hypotheses to data.** The likelihood $p(X|C)$ and the prediction $\mathbb{1}\left[X \in C\right]$ both require comparing natural language $C$ against a nonlinguistic datum $X$. For the domains we consider it is convenient to implement this comparison by translating the concept $C$ into Python using an LLM—which we do deterministically, via greedy decoding—and then execute the Python program on $X$, similar to [41, 42]. We do not view this latent program as essential to the general framework: Alternatively, another neural network could serve to link $X$ to $C$, which would allow fuzzier comparisons between hypotheses and data. Fig. 1 diagrams these details as a Bayesian network.

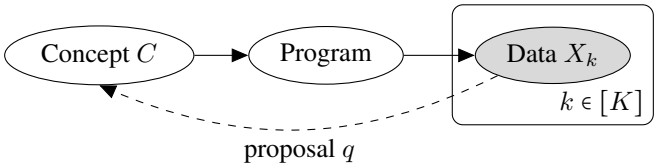

Figure 1: Model as Bayesian network. The program is a deterministic function of $C$.

## 4 The Number Game

The Number Game is a few-shot concept learning setup covered by classic textbooks and dissertations [5, 43]. Participants playing The Number Game are given a few example numbers belonging to a hidden concept, and then rate how likely it is that other numbers also belong to the same concept. Given just the example number *16*, the concept could be 'square numbers', 'powers of two', 'evens', 'odds but also 16', '97 and 16', 'numbers ending in 6', or infinitely many other concepts. With more examples such as *16, 8, 2, 64*, humans consistently rate powers of two as almost certainly in the concept, but gradations of uncertainty remain: other evens like 24 could plausibly belong in the concept, but humans rate odds like 23 as extremely unlikely. Examining human judgments for these and other concepts reveal a variety of belief states, including sharp, all-or-none concepts like 'powers of two,' but also soft graded concepts like 'numbers around 20' (Fig. 2, blue bars).

**Human Data.** We take human data from [43]. Eight human participants rated test numbers on a scale of 1-7, given different training sets of example numbers. We model the average human rating for each test number, on each training set of example numbers.

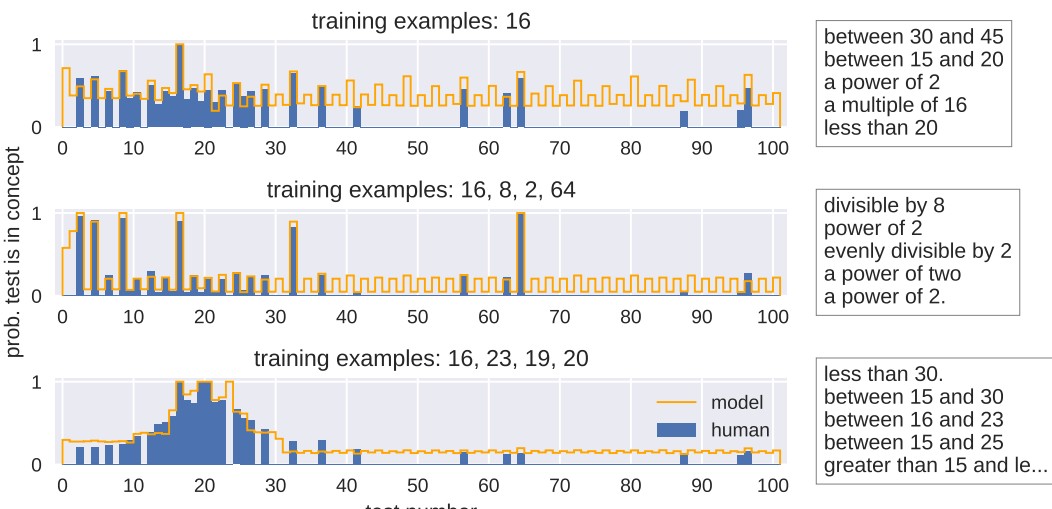

Figure 2: Number Game human judgments (blue bars) for different example data. For instance, the top plot shows that after seeing that *16* belongs to the concept, humans rate *64* as around 50% likely to belong to that same concept. Bars at zero correspond to missing data. Orange curves show our model's predictions. The text to the right of each plot shows 5 samples from our model's approximate posterior after proposing 100 concepts using $q$. Human data from [43]. See also Appendix Fig. 7

**Prior.** We consider two different prior distributions. The **pretrained prior** scores the log likelihood of each concept $C$ using an open source language model, specifically, CodeGen 350M [44]. The **tuned prior** first extracts features of $C$ using a pretrained sentence feature extractor $\phi$, specifically all-MiniLM-L6 [45], which outputs a 384-dimensional feature vector. The tuned prior maps those features to an unnormalized log probability via a linear layer with parameters $\theta$:

$$\text{Tuned prior:} \quad p_\theta(C) \propto \exp\left(\theta \cdot \phi(C)\right) \tag{6}$$

**Likelihood.** Evaluating $p(X|C)$ requires first determining which numbers belong to the concept $C$. To efficiently enumerate those numbers, we translate $C$ from natural language to Python using Codex code-davinci-002 [23], a language model trained on source code. We run the Python code on the numbers 1..100 to determine the members of $C$. Given the members of $C$, we assume numbers are sampled uniformly from $C$ with probability $(1 - \epsilon)$, and otherwise sampled uniformly from 1..100:

$$p(X|C) = (1 - \epsilon)\frac{\mathbb{1}\left[X \in C\right]}{|C|} + \epsilon\frac{1}{100} \tag{7}$$

**Parameter fitting.** We want a single parameter setting that works for *all* of the training example sets, so we fit the above parameters to the average human judgment for each test number, and for each set of examples. When comparing model predictions against the average human judgment on a particular test number, we always holdout that particular human data from the parameter fitting. We use Adam [46] to perform maximum likelihood estimation of the parameters, following Eq. 5.

**Temperature, Platt transform.** Because human subjects rated on a scale of 1-7, we introduce a learnable Platt transform between the model's predicted probabilities and the human judgments [47]. We also place a learnable temperature parameter on the posterior.

**Proposal distribution.** We implement $q$ using Codex code-davinci-002 [23]. We prompt Codex by adapting the cover story given to the human subjects, then append the training example numbers $X_{1:K}$ and have it complete the natural language description of the hidden concept.

**Alternative models.** We compare against GPT-4 to understand the few-shot learning abilities of a bleeding-edge large language model. Prompted with example numbers belonging to a concept, together with a test number, we measure the probability that GPT-4 responds "yes" to the test number belonging to the concept, vs responding "no". We fit a Platt transform to those probabilities. We also compare against DreamCoder [21], a recent Bayesian Program Learning system. Starting with a Number Game DSL, DreamCoder learns a prior and trains a neural proposal distribution. DreamCoder considers $10^4$ hypotheses at test time—two orders of magnitude more than our model—and trains its neural network on $10^5$ synthetic number concepts ("dreams"). We further contrast against Latent Language [31], using the same Codex-based proposal and likelihood distributions, and the same learnable Platt transform. Finally, we consider versions of our model that encode hypotheses in Python instead of natural language ('code prior'), as well as a version of our model that ablates the proposal distribution by generating concepts unconditioned on $X_{1:K}$ ('no proposal dist.').

**Results.** Fig. 3 shows that an out-of-the-box pretrained natural language prior offers a decent fit to the human data after proposing 50-100 hypotheses. Our tuned prior achieves a very close fit to the human data, again using 50-100 samples. Switching from English to Python significantly degrades model fit, even when the Python prior is learned ('tuned code prior'). Ablating the proposal distribution–sampling hypotheses from the prior–also provides a poor fit: the space of possible number hypotheses is too vast to be randomly guessed without looking at the data.

These results establish that a language-generating proposal distribution can support efficient inference, and accurately model the average of a small pool of human participants. Recall that we are modeling the average of 8 human judgements, and a good fit to this average requires 50-100 samples. Therefore, our model suggests that each human might only need to draw a couple dozen samples, which we think is psychologically plausible (and practical, from an engineering perspective). In contrast, the original Number Game model considered over 5000 hypotheses [43], and a typical BPL system such as DreamCoder still lags our model, even when it considers orders of magnitude more hypotheses.

Last, GPT-4's judgments are decidedly different from humans. This does not mean that GPT-4 is wrong in its predictions: there is no ground-truth for whether the number 30 should belong to the same concept as the number 60. To better understand how humans and models stack up against each other, we next consider a richer domain where accuracy can be more objectively measured.

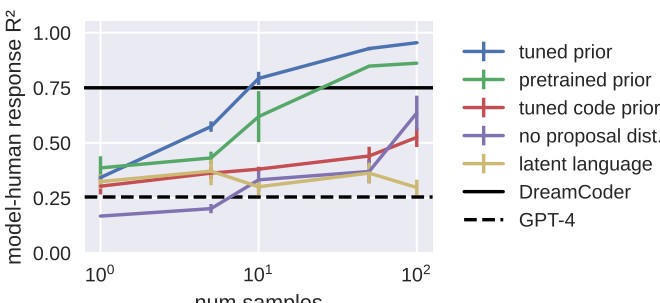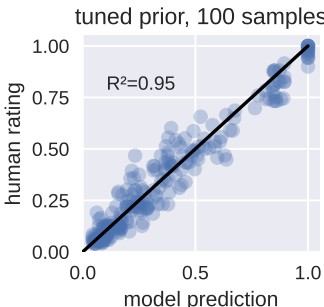

Figure 3: How well different models predict held-out human judgments ($R^2$), as a function of the sampling budget (left panel, X-axis, log scale). Error bars: ±SEM over 3 runs with different seeds. Variance across runs decreases with number of samples. See Appendix Fig. 8-9 for further results.

## 5 Logical Concepts

We next consider concepts with more complex logical structure. Consider a concept such as *bachelor*, defined as "unmarried man", or *Valedictorian*, defined as "the person, within a single school, with the highest GPA". Using the primitives of propositional logic, we can define *bachelor*: $(\text{Male} \wedge \neg \text{Married})$. Using the more expressive language of first-order logic, which includes quantifiers, we can define *valedictorian* as $\text{Valedictorian}(x) \iff (\forall y : \text{School}(x) = \text{School}(y) \implies \text{GPA}(x) \geq \text{GPA}(y))$. Discovering the discrete logical structure that best explains a dataset of examples is a well-known AI challenge [48, 49, 50]. Within the cognitive sciences, understanding how people come to grasp logical relations has been proposed to be a key component of understanding how people comprehend number systems, geometry, causal processes, social and kinship relations, and other domains [51, 52, 53, 54].

For our modeling, we consider an online learning setup from [55] where a learner observes a stream of examples of a unknown logical concept. On each example, the learner observes a fresh batch of 1-5 objects, and must pick out which objects belong to the hidden concept. The learner then gets feedback on which objects in the batch actually belonged to the concept, and then the process repeats for a new batch. Fig. 4 illustrates this experimental setup: each object is a shape defined by its size (small, medium, large), color (green, yellow, blue), and shape (triangle, rectangle, circle). Recording each human response to each shape on each batch gives a fine-grained learning curve capturing how learning unfolds over dozens of examples. These learning curves signal what concepts people readily learn, what patterns of mistakes they make, and what concepts remain essentially unlearnable from even dozens of examples. We obtain this human data from [55], which covers 112 concepts, collecting judgements from 1,596 human participants as they attempt to learn each concept over 25 batches of examples. These 25 batches corresponds to ≈ 75 examples/concept, and each concept run on ≈ 20 participants. Most human learning happens over the first dozen batches, so we take the first 15/25 batches, which conveniently also nearly halves the cost and time of running the model.

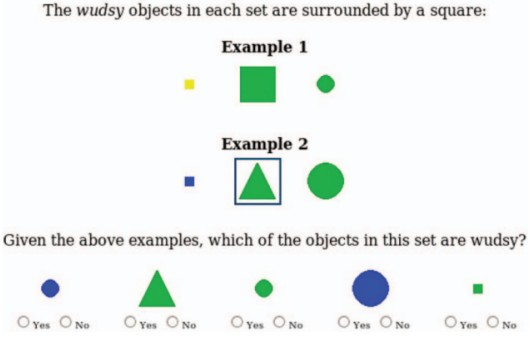

Figure 4: Concept learning experiment from [55] (illustration used with permission). On each batch, participants label which shapes they think belong to a new concept (called *wudsy*). Previous batches are shown with the ground truth positive examples surrounded by a square. From these examples, participants might infer a simple concept like "green triangles", and select the second test object.

**Model.** Our modeling approach is similar to The Number Game, except we now have a discriminative learning problem instead of a generative one, and an online learning setup where the learner observes a stream of examples. To model online learning, we draw fresh proposals from $q$ for each training batch, and perform Bayesian inference over all proposals drawn so far. To model discriminative learning, each example is now a triple $(B, T, Y)$, where $B$ is a batch of shapes, $T$ is a test shape in that batch, and $Y$ is zero or one depending on whether $T$ in $B$ is an example of the concept.

Our likelihood model assumes human subjects predict according to $C$ with probability $(1 - \epsilon)$ and otherwise predict randomly with a base rate $\alpha$ of labeling an example as positive. Following [55, 56] we model a memory decay process where the relative importance of earlier observations falls off according to a power law with parameter $\beta$:

$$p(Y = 1 | B, T, C) = (1 - \epsilon)\mathbb{1}\left[(B, T) \in C\right] + \epsilon\alpha \tag{8}$$

$$\log p(X_{1:K} | C) = \sum_{(B_k, T_k, Y_k) \in X_{1:K}} (1 + K - k)^{-\beta} \log p(Y | B, T, C) \tag{9}$$

As before, we translate hypotheses from natural language into Python using code-davinci-002, and evaluate the likelihood term above by running Python code. We consider both pretrained and tuned priors. Our proposal distribution $q$ comes from running GPT-4 on prompts that illustrate previous batches, either as truth tables (for propositional concepts) or as a raw list of previous observed batches (for higher-order concepts). We again place a learnable temperature on the posterior.

**Bayesian Program Learning Baseline (BPL).** We contrast with a strong BPL baseline. It uses a grammar over expressions in first-order logic, plus predicates for shape, color, and size, totaling 28 primitives that were selected by the creator of the logical concept learning dataset (A.3.3). The BPL baseline uses the same memory-decay likelihood model, and fits (tunes) its prior by estimating probabilities for each logical primitive. It is implemented in Fleet [57], the state-of-the-art in fast parallelized MCMC over grammatically structured hypothesis spaces.

Our model differs in two important ways. First, the baseline is given first-order primitives that were chosen specifically for this dataset. While our model can use natural language expressing first-order concepts (e.g., *the only* for ∃!), it can also express concepts like *objects with the least common color* that are unrepresentable by the baseline, and which are not in the dataset.

The second difference is that our model supports efficient inference via bottom-up proposals, while this baseline performs a stochastic search over hypotheses (MCMC), requiring many more samples. It takes $10^6$ Metropolis-Hastings proposals per batch, and per learning curve, totaling $\approx 10^9$ proposals, which are deduplicated to yield $\approx 45,000$ unique concepts, which provide the support of subsequent posterior inference. This means every BPL posterior is informed by the total $\approx 10^9$ sampling moves.

**Results.** Our model's predictions generally align well with human judgments (Fig. 5). Using 100 proposals per batch, our model explains 81% of the variance in human responses ($R^2 = .81$), which is much higher than GPT-4 on its own. The model is also more accurate at the actual task than GPT-4, and within 3% of human-level (Fig. 6C). The model also fits the human data somewhat better than the BPL baseline, even when that baseline is granted an exorbitant sampling budget. Tuning the prior proves critical, possibly because these logical concepts are more complex than the earlier number concepts, and so can be expressed in a wider variety of syntactic forms. The pretrained model is highly sensitive to syntax and cannot learn to attend to semantic features, unlike the tuned model.

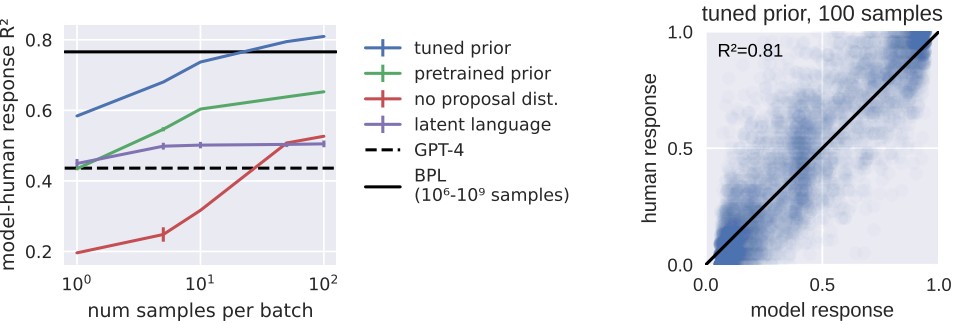

Figure 5: Model fits on holdout data. Error bars: ±SEM over 3 runs. (Error bars often close to zero)

Although our model explains most of the variation in the human responses, nontrivial variation remains. One possible reason is that we are modeling the responses of many human subjects, and different subject groups per concept, so it is difficult to capture the full variability of the human responses: Building a model that accurately reflects the judgments of a population of humans may be much more difficult, and require a higher sampling budget, than building a model of a single person. The model also slightly underperforms humans, and so a higher fidelity model might come from simply performing the task better. (Although one can slightly surpass human accuracy by optimizing purely for task performance, doing so degrades model fit. See Fig. 6C, 'tune for accuracy'.) More fundamentally, the data come from many successive trials, so they likely contain memory and order effects such as garden-pathing [58] and anchoring [39] which are not well accounted for by the pure probabilistic inference our model approximates, nor by our simple memory decay model.

At the same time, our model does account for many fine-grained details in the behavioral data, including predicting specific patterns of successes and failures (Fig. 6A). Because our approach is manifestly interpretable—it explicitly verbalizes its hypotheses in human language—we can inspect its maximum a posteriori concept at the end of each learning episode, and observe that its successes typically occur because its verbalization of the concept describes the correct first-order law. Conversely, when humans make highly selective failures, we can probe the model to suggest what alternative hypotheses humans may have incorrectly inferred (Fig. 6A, top right).

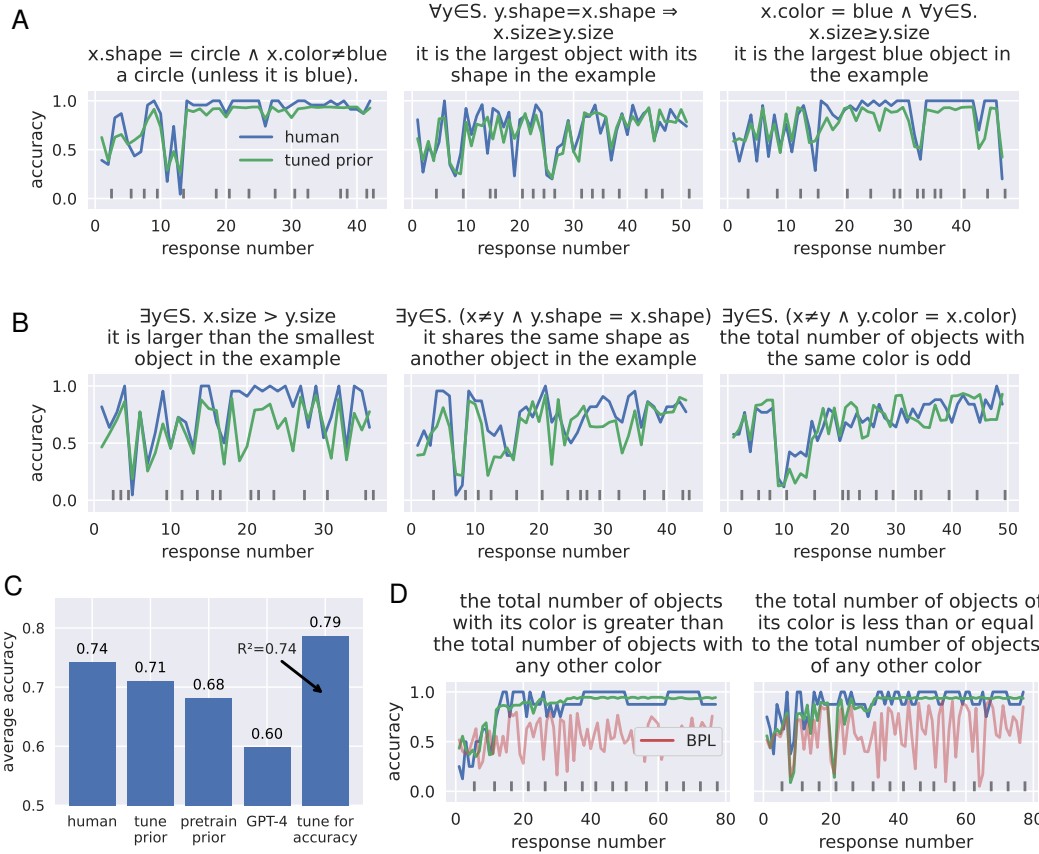

Figure 6: **A.** Left and middle: learning concepts isomorphic to *bachelor* and *valedictorian*. Dark ticks delimit batches. Above each plot is the ground-truth logical expression and the predicted natural language description. Right: the model can explain human mistakes by verbalizing what other concept humans might have been misled into learning. Both humans and the model seem to have learned *largest blue* instead of *the largest and it also happens to be blue*. **B.** Further learning curves, including somewhat worse fits, and getting the right answer despite having a slightly odd solution (rightmost). **C.** The model is close to human performance after extracting a prior from human judgments. **D.** New concepts run on new participants. All results on holdout learning curves.

**Modeling new concepts.** To test the flexibility of our model, we created two new concepts for evaluating both the model and a new group of human participants. These two concepts were *shapes with the majority color*, and *shapes with the least common color*. 16 human subjects participated in an IRB-approved study (A.2). Our study finds that humans rapidly learn these concepts (Fig. 6D.)

We also test our model on these new concepts, but using the prior estimated earlier on the data in [59]. Our model correctly predicts that humans will learn the concept of *majority color* after just 2 batches of examples. It predicts learning *minority color* after 4 batches, while humans need just 3, but the primary result—that this concept is very learnable—holds for both humans and the model.

Although these two new concepts are easy to explain in words, and could be expressed in first-order logic with the right primitives—set cardinality and quantification over colors—neither concepts are learnable by the BPL baseline. This is because both concepts are simply unrepresentable: despite being equipped with 28 primitives, those primitives were designed without anticipating these new concepts. This highlights the fact that it is difficult to preprogram a sufficiently broad set of primitives that can efficiently encode all the different concepts people might learn.

# 6 Discussion

**Putting humanlike inductive biases into machines.** Humans excel at rapidly mastering new tasks and understanding new concepts in part because they have a good inductive bias or prior [60, 61]. Imparting similar priors upon machines is therefore an important problem. Our work gives a recipe for training such a prior by fitting it directly to human judgments, marginalizing out the natural language, meaning we never need to elicit natural language from human subjects. Because our approach simply tunes a small network on top of a large open-source pretrained model, and because it can be fit to raw human judgments, we hope that it can serve as a broadly applicable engineering strategy for extracting and using human priors. Recent work has also explored complimentary strategies for instilling a humanlike prior upon a machine learning system. For example, Kumar et al. 2022 [9] show that training neural networks with auxiliary linguistic tasks, and with auxiliary programming tasks, causes their representations to better align with human priors.

**Bayesian Program Learning (BPL).** Our model has similar motivations to BPL. Because we translate the natural language into Python to compute the likelihood, it is possible to see our approach as BPL with an unusual prior: $p(\text{program}) = \sum_{\text{NL}} p(\text{NL})p(\text{program}|\text{NL})$. In that sense, our work provides a new prior for BPL, together with the demonstration that it is possible and practical to do BPL over a Turing-complete language like Python. Relatedly, recent BPL modeling has found synergies between natural language and program representations for cognitive AI models [9, 62].

Beyond BPL, combining probabilistic reasoning with expressive symbolic representations has long been an appealing paradigm [63, 64, 65], although the expressivity of the symbolic language must be balanced against the tractability of inference [66, 67]. Guiding inference with a neural model is a natural choice, but this is hard because of the lack of natural training data (though synthetic data can help: [68, 69, 70]). Encoding knowledge in natural language allows pretrained neural models to guide inference, and it could be fruitful to examine statistical-relational AI [71] in light of that fact.

**Large Language Models.** Our work suggests that an out-of-the-box large language model is not an effective approach to inductive reasoning, at least on its own. Bayesian mechanics are needed to dampen the unpredictability of the language model. To the extent that being Bayesian is a normative account of rational behavior, our work offers a framework for enabling language models to draw more rational inferences, and ultimately generalize in more predictable and human-like ways by tuning their prior beliefs to match human data.

**The Language of Thought.** Our work connects to the Language of Thought Hypothesis [72], which says that human learning and thinking relies on an inner symbolic language. This has been a productive framework for computational modeling [15, 16, 18, 73]. In its most literal forms, language-of-thought models are afflicted by the *curse of a compositional mind* (Spelke 2022 [74]): the free-form recombination of concepts yields a combinatorial explosion, which here we address by using pretrained knowledge from language to guide the learner. Whatever the true Language of Thought looks like, however, it must have a wide array of composable basic concepts in order to explain the breadth, flexibility, and generality of human thinking. Natural language, even if it is not actually the same as our inner mental language, acts as a vast reservoir of human concepts,

and provides a flexible algebra for combining them. Therefore a reasonable near-term strategy for modeling human thinking may be to use natural language as a heuristic approximation to an inner Language of Thought, despite strong evidence that human thinking need not engage language-processing brain regions [75].

**Rational Process Modeling.** As a rational process model, our account of concept learning bridges the computational and algorithmic levels of the Marr hierarchy [28, 29]: We commit to a Bayesian computational-level theory, and a particular Monte Carlo algorithm as a rational approximation to that theory. One important sense in which our account is inadequate is that we do not actually explain how the bottom-up process works, or how it came to be learned. We merely require that it is stochastic and unreliable, but occasionally correct enough to not need many millions of proposals. Given those requirements, a modern large language model is a reasonable surrogate for this bottom-up process, even if it its inner workings might differ greatly from human bottom-up proposal processes.

**Generalizability of the theoretical framework.** The basics of the model make few commitments, yet instantiating it requires selecting specific language models, engineering prompts and likelihoods, etc. More broadly, a high-resolution cognitive model, particularly a structured Bayesian one, requires domain-specific modeling choices. How much credit should we assign to the general theoretical framing, as opposed to particular engineering decisions? Although our paradigm introduces new degrees of freedom, such as which LLMs/prompts to use, it removes others, such as the grammatical structure of the symbolic hypothesis space. On balance, we are cautiously optimistic that the framework will generalize with less domain-specific tinkering, at least for abstract symbolic domains, because it replaces hand-designed symbolic hypothesis spaces with pretrained neural models, and because reasonable "default" neural networks worked well across our experiments.

**Inference, Natural Language, and Recursive Reasoning.** There is a rich literature on natural language and probabilistic inference from a social or communicative perspective, such as modeling pragmatic inference [76, 77, 78, 79]. These models often use a recursive reasoning formulation where each agent represents the beliefs of other agents, including beliefs about themselves, and so on, recursively. Our work does not consider the communicative function of natural language, and so does not engage with these issues. However, models of pedagogical inductive learning—where a helpful teacher selects the training examples—benefit from similar forms of recursive reasoning, because the teacher and learner must reason about each other's beliefs [80]. These insights could be important for building versions of our model that might simulate aspects of how humans learn from each other.

**Limitations.** Our model performs induction via discrete structure learning. Given the combinatorial difficulty of structure learning, it is unclear whether our approach can scale to inferring complex systems of symbolic rules. We believe recent work on iteratively refining language model outputs may be promising here [81, 82]. The present form of our model is also limited to processing discrete symbolic input-outputs. Actual human thinking connects with the messy perceptual world. It would be valuable to understand whether this limitation can be addressed using multimodal language models [83], or approaches that interface separate language and vision modules [84, 85]. It is worth noting that BPL models can straightforwardly interoperate with perceptual data [10, 16, 18], and that many outstanding challenge problems within AI have at their core a perceptual-reasoning process, such as Bongard problems [86] and the Abstraction and Reasoning Corpus [1].

Currently, the model relies on costly, energy-intensive models for its proposal distribution, a constraint that might be mitigated by open-source models and network compression [87].

Last, a strong justification for formal representations is that they allow specifying knowledge precisely and unambiguously [88]. Natural language is usually imprecise and ambiguous, which we deferred addressing by translating language into Python. It remains to be seen whether language models can be coaxed into producing sufficiently precise language to support representing knowledge solely in natural language, or if refinement into precise languages like Python offers the better scaling route.

**Code and data available at:** https://github.com/ellisk42/humanlike_fewshot_learning

**Acknowledgements.** We are grateful to Steven Piantadosi for providing the raw human data and Fleet results for the logical concepts, as well as Joshua Tenenbaum for providing his Number Game data, and Mathias Sablé-Meyer for comments on the manuscript and work.

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

# A  Appendix

## A.1  Supplemental Results

Fig. 7 illustrates model predictions across every Number Game concept in [43].

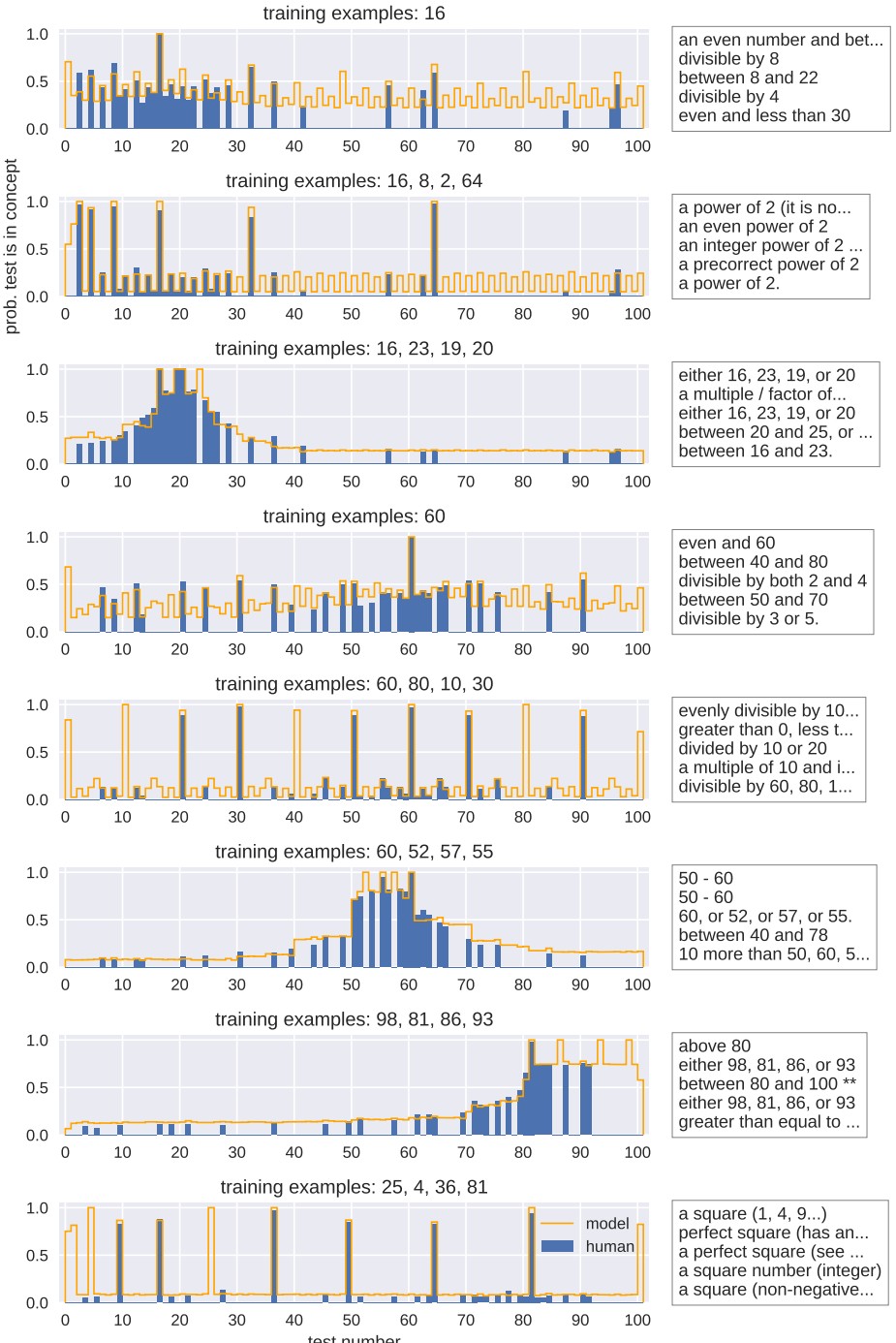

Figure 7: Model predictions across every Number Game concept in [43], using 1000 $q$ samples

Recall that we deduplicated the proposals instead of performing actual importance sampling. Fig. 8 contrasts model fit for importance sampling and deduplication. We originally did deduplication simply because importance sampling is not possible with GPT-4, and GPT-4 proved necessary for the logical concepts. On number concepts we used code-davinci-002, from which we can construct an importance sampler because it exposes the log probability of its samples. On number concepts deduplication provides a fit that is on-par (actually slightly better) compared to importance sampling (Fig. 8).

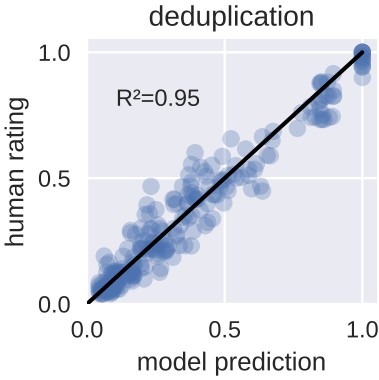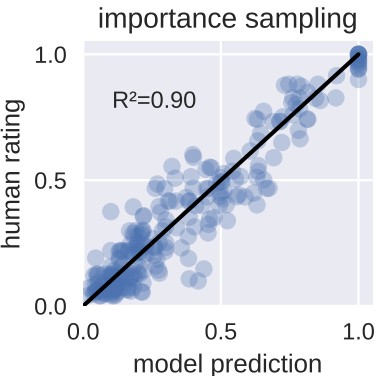

Figure 8: Monte Carlo inference using deduplication instead of importance sampling does not harm model fit to human data. The above figures show Number Game models using a learned prior and 100 samples, and show predictions only on holdout data.

We also studied the effect of replacing the closed-source code-davinci-002 (Codex) with the open-source 70B LLaMA2 [89]. This helps in understanding if open LLMs can be used to build models like ours. Additionally, as the open models are presumably weaker, swapping out Codex for LLaMA2 serves as a way of softly ablating either the likelihood or proposal distribution, both of which use an LLM. We find that the (presumably weaker) LLaMA2 can substitute fully as a likelihood. Using LLaMA2 as a proposal distribution impairs model fit in the low-sample regime (Fig. 9), but with enough samples, the weaker model comes close to 'catching up'. This highlights the fact that an explicitly Bayesian reasoning process can compensate for deficiencies in the LLM, given enough Monte Carlo samples.

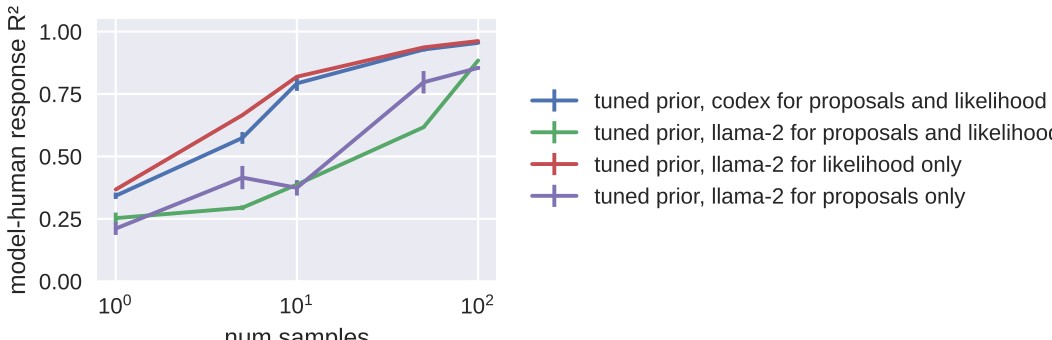

Figure 9: Comparing versions of our model that swap out the closed-source Codex model for the open-source LLaMA2 70B model, for both the likelihood and the proposal distributions.

## A.2 Human Study

16 participants were recruited primarily through a Slack message sent to a channel populated by members of our academic department. Participants had an average age 28.1 (stddev 13.6, all over

18), and were 7 male/5 female/1 nonbinary/3 declined to answer. Participants were randomly split between the concepts of *most common color / least common color*. Each participant went through 15 trials, and took an average of 294s to complete those 15 trials. In exchange for participating in the study, participants received $10 in Amazon gift cards. Fig. 10 illustrates the web interface shown to our human participants, including the cover story.

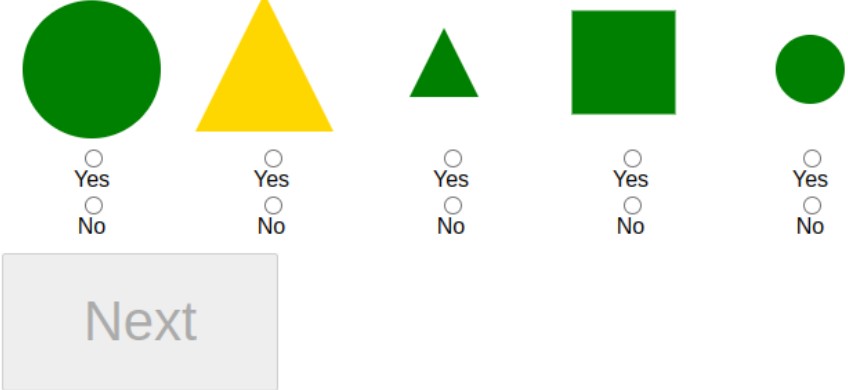

Figure 10: Cover story and web interface for our version of the logical concept learning study, which is based on [59]

### A.3  Modeling

#### A.3.1  Temperature and Platt Transform

Adding a temperature parameter $T$ to a model corresponds to computing the posterior via

$$p_{\text{Temp}}(X_{\text{test}} \in C | X_{1:K}) \approx \sum_{C \in \{C^{(1)}, \dots, C^{(S)}\}} w^{(C)} \mathbb{1}\left[X_{\text{test}} \in C\right], \text{ where}$$

$$w^{(C)} = \frac{\left(\tilde{w}^{(C)}\right)^{1/T}}{\sum_{C'} \left(\tilde{w}^{(C')}\right)^{1/T}} \text{ and } \tilde{w}^{(C)} = p(C)p(X_{1:K}|C)\mathbb{1}\left[C \in \{C^{(1)}, \dots, C^{(S)}\}\right]$$

$$(10)$$

Adjusting the predictions of a model using a Platt transform corresponds to introducing parameters $a$ and $b$ which transform the predictions according to

$$p_{\text{Platt}}(X_{\text{test}} \in C | X_{1:K}) = \text{Logistic}(b + a \times \text{Logistic}^{-1}\left(p(X_{\text{test}} \in C | X_{1:K})\right)) \qquad (11)$$

For the number game, every model has its outputs transformed by a learned Platt transform. This is because we are modeling human ratings instead of human responses. We expect that the ratings correspond to some monotonic transformation of the human's subjective probability estimates, and so this transformation gives some extra flexibility by inferring the correspondence between probabilities and ratings. Logical concept models do not use Platt transforms.

### A.3.2   Parameter fitting

Training consists of fitting the parameters $T$, $\theta$ (for the prior), $\epsilon$ (for the likelihood), $\alpha$ and $\beta$ (for the logical concepts likelihood), and Platt transform parameters $a$, $b$ (for the Number Game). In practice, this amounts to around 400 parameters, almost all of which come from $\theta$.

We fit these parameters using Adam with a learning rate of 0.001. We perform 1000 epochs of training for the Number Game, and 100 epochs for logical concepts. There is a tenfold difference in the number of concepts, so this way they take about the same number of gradient steps.

For the number game we do 10-fold cross validation to calculate holdout predictions. For logical concepts we use the train-test split introduced in [59], which involves running different groups of human subjects on each concept twice, with different random examples. One sequence of random examples is arbitrarily designated as training data, and the other as holdout data.

All model were trained on a laptop using no GPUs. Training takes between a few minutes and an hour, depending on the domain and the model.

Some of the parameters that we fit, namely $\epsilon$, $\alpha$, $\beta$, cannot be negative. To enforce this we actually optimize the inverse logistic of those parameters.

### A.3.3   MCMC over Logical Expressions

Fleet was used[1] to perform MCMC over logical expressions with the domain-specific primitives in this file, which include:

$$\text{true, false : boolean}$$
$$\text{blue, yellow, green : object} \rightarrow \text{boolean}$$
$$\text{rectangle, circle, triangle : object} \rightarrow \text{boolean}$$
$$\text{small, medium, large : object} \rightarrow \text{boolean}$$
$$\text{and, or, } \iff, \implies : \text{boolean} \times \text{boolean} \rightarrow \text{boolean}$$
$$\forall, \exists : (\text{shape} \rightarrow \text{boolean}) \times 2^{\text{object}} \rightarrow \text{boolean}$$
$$\text{filter} : (\text{object} \rightarrow \text{boolean}) \times 2^{\text{object}} \rightarrow 2^{\text{object}}$$
$$\epsilon : \text{object} \times 2^{\text{object}} \rightarrow \text{boolean}$$
$$\iota : 2^{\text{object}} \rightarrow \text{object} \cup \{\bot\}, \text{ unique set element}$$
$$\text{empty} : 2^{\text{object}} \rightarrow \text{boolean}$$
$$\text{same\_shape, same\_color, same\_size : object} \times \text{object} \rightarrow \text{boolean}$$
$$\text{size<, size≤, size>, size≥, : object} \times \text{object} \rightarrow \text{boolean}$$

The model first constructed a large hypothesis space by performing MCMC for 1 minute per batch, and per learning curve. In one minute, Fleet makes approximately $10^6$ MH proposals. There are a little more than 200 learning curves, and 25 batches per curve, for a total of about 5 billion MCMC proposals. In the main text, we abbreviate this analysis by referring to $10^9$ proposals.

The top 10 samples per batch and per learning curve were retained. These top 10 samples samples were then deduplicated to yield 45 thousand hypotheses. Parameter fitting and posterior estimation was performed solely over those 45 thousand hypotheses.

Quantitatively, these are vastly more proposals than the models introduced in this paper. Quantitatively, these proposals are also derived in a very different way: the hypothesis space for the BPL learner is

---

[1]Running the model was graciously performed by the authors of [59], who provided us with the raw data.

actually informed by data on other learning curves, and also on the same learning curve, but in the future batches.

It is in this sense that the BPL model is a computational-level theory, and not a process model, because human subjects could not be proposing hypotheses using data that is going to be seen in the future, or on other learning curves. However, the above strategy for proposing hypotheses is a very reasonable heuristic for constructing the support of the space of plausible logical hypotheses that a human learner might ever think of.

### A.3.4   DreamCoder baseline

DreamCoder was initialized with the following domain-specific primitives:

$$1, 2, 3, ..., 100 : \text{int}$$
$$\text{squares} : 2^{\text{int}}$$
$$\text{singleton} : \text{int} \to 2^{\text{int}}$$
$$\text{interval} : \text{int} \times \text{int} \to 2^{\text{int}}$$
$$\text{intersect}, \text{union} : 2^{\text{int}} \times 2^{\text{int}} \to 2^{\text{int}}$$
$$\text{powersof}, \text{multiplesof} : \text{int} \to 2^{\text{int}}$$

Each of the 8 stimuli from [43] was converted into a single task. The discrete structure of the prior was held fixed because there was no learnable common structure across the 8 tasks, but the continuous parameters of the prior were updated at each wake-sleep cycle. We estimated those parameters from the topK=100 highest posterior programs discovered for each task at each wake-sleep cycle. The neural recognition model took as input a 100-dimensional binary vector encoding the support of a number concept, which was then processed by an MLP with 64 hidden units. The recognition model was trained on a 50/50 split of replays/dreams for approximately $10^5$ training examples per wake-sleep cycle. During the waking phase, program synthesis was provided with a timeout of 60 seconds per task per cycle, enumerating approximately $10^4$ programs per task.

We confirmed by manual inspection that high-likelihood programs were discovered for of the eight tasks. Therefore the difference between DreamCoder and our model cannot be attributed to a failure on the part of DreamCoder to discover valid programs. Instead, we attribute the difference to differences of representation (natural language vs lambda calculus), together with differences in the ways that parameters are estimated (maximizing the marginal likelihood of the tasks, vs directly fitting human judgments).

### A.4   Prompting

### A.4.1   Proposing hypotheses

For the number game we use the following prompt for code-davinci-002 to generate candidate concepts in natural language, given examples $X_{1:K}$. The example number concepts given in the prompt come from the cover score given to human participants [43]:

```
# Python 3
# Here are a few example number concepts:
# -- The number is even
# -- The number is between 30 and 45
# -- The number is a power of 3
# -- The number is less than 10
#
# Here are some random examples of numbers belonging to a different ↗
    ↳ number concept:
# X_{1:K}
# The above are examples of the following number concept:
# -- The number is
```

where $X_{1:K}$ is formatted by listing the numbers with a comma and a space between them.

For the number game we used the following prompt to generate candidate concepts in python (code baseline):

```
# Python 3
# Here are a few example number concepts:
# -- The number is even
# -- The number is between 30 and 45
# -- The number is a power of 3
# -- The number is less than 10
#
# Here are some random examples of numbers belonging to a different ↙
    ↳ number concept:
# $X_{1:K}$
# Write a python function that returns true if `num` belongs to ↙
    ↳ this number concept.
def check_if_in_concept(num):
    return
```

For logical concepts we used the following few-shot prompt for GPT-4 to generate candidate concepts:

```
Here three simple concepts, which specify when an object is ↙
    ↳ 'positive' relative to an example collection of other ↙
    ↳ objects. Before giving the rule for each concept, we give ↙
    ↳ examples of collections of objects, and which objects in the ↙
    ↳ collection are 'positive'.

Concept #1:
    An Example of Concept #1:
        POSITIVES: (big yellow rectangle)
        NEGATIVES: (big green circle), (medium yellow rectangle)
    Another Example of Concept #1:
        POSITIVES: (medium yellow rectangle)
        NEGATIVES: (big red circle), (small green circle)
Rule for Concept #1: Something is positive if it is the biggest ↙
    ↳ yellow object in the example.

Concept #2:
    An Example of Concept #2:
        POSITIVES: (small yellow circle), (medium yellow rectangle)
        NEGATIVES: (big green circle), (big blue rectangle)
    Another Example of Concept #2:
        POSITIVES: (big blue circle), (medium blue rectangle)
        NEGATIVES: (small green circle), (medium yellow rectangle),
Rule for Concept #2: Something is positive if there is another ↙
    ↳ object with the same color in the example.

Concept #3:
    An Example of Concept #3:
        POSITIVES: (small yellow circle), (medium yellow rectangle)
        NEGATIVES: (big green circle), (big blue rectangle)
    Another Example of Concept #3:
        POSITIVES: (small blue circle), (small blue triangle), ↙
            ↳ (medium blue rectangle)
        NEGATIVES: (medium green triangle), (big yellow rectangle)
    Another Example of Concept #3:
        POSITIVES: (big red rectangle), (medium red rectangle), ↙
            ↳ (big red triangle)
        NEGATIVES: (medium green triangle), (big yellow rectangle)
Rule for Concept #3: Something is positive if it is the same color ↙
    ↳ as the smallest triangle in the example.

Now here are some examples of another concept called Concept #4, ↙
    ↳ but this time we don't know the rule. Infer ten different ↙
    ↳ possible rules, and make those ten rules as simple and ↙
    ↳ general as you can. Your simple general rules might talk ↙
```

```
        ↳ about shapes, colors, and sizes, and might make comparisons ↙
        ↳ between these features within a single example, but it ↙
        ↳ doesn't have to. Remember that a rule should say when ↙
        ↳ something is positive, and should mention the other objects ↙
        ↳ in the example, and should be consisting with what you see ↙
        ↳ below.

Concept #4:
    X_{1:K}
Rule for Concept #4: Something is positive if...

Now make a numbered list of 10 possible rules for Concept #4. Start ↙
    ↳ by writing "1. Something is positive if". End each line with ↙
    ↳ a period.
```

Each sample from the above prompt generates 10 possible concepts formatted as a numbered list. We draw 10 times at temperature=1 to construct 100 hypotheses. To obtain fewer than 100 hypotheses we take hypotheses from each sampled list in round-robin fashion. We found that asking it to generate a list of hypotheses generated greater diversity without sacrificing quality, compared to repeatedly sampling a single hypothesis.

The above prompt provides in-context examples of first-order rules. We also tried using a different prompt for propositional concepts that illustrated the examples as a truth table, and gave in-context example rules that were propositional:

```
Here are some example concepts defined by a logical rule:

Rule: a triangle.
Rule: a green rectangle.
Rule: big or a rectangle (unless that rectangle is blue).
Rule: not both big and green.
Rule: either big or green, but not both.
Rule: either a rectangle or not yellow.
Rule: a circle.

Now please produce a logical rule for a new concept. Your rule ↙
    ↳ should be consistent with the following examples. It must be ↙
    ↳ true of all the positive examples, and not true of all the ↙
    ↳ negative examples. The examples are organized into a table ↙
    ↳ with one column for each feature (size, color, shape):

X_{1:K}

Please produce a simple rule that is consistent with the above ↙
    ↳ table. Make your rule as SHORT, SIMPLE, and GENERAL as ↙
    ↳ possible. Do NOT make it more complicated than it has to be, ↙
    ↳ or refer to features that you absolutely do not have to refer ↙
    ↳ to. Begin by writing "Rule: " and then the rule, followed by ↙
    ↳ a period.
```

Using the first order prompt for every concept gives a $R^2$ = .80 fit to the human responses. Using both prompts gives the $R^2$ = .81 result in the main paper: the propositional prompt for the propositional problems, and the first order prompt for the higher order problems. We strongly suspect that a single prompt that just showed both propositional and higher-order in-context examples would work equally well, given that a single first-order prompt works about as well also, but we did not try that it would have required rerunning all of our GPT-4 queries, which would have had high cost.

On the first batch, the learner has not observed any examples. Therefore the above prompts do not apply, and we use a different prompt to construct an initial hypothesis space:

```
Here are some example concepts defined by a logical rule:

Rule: color is purple.
Rule: shape is not a hexagon.
```

```
Rule: color is purple and size is small.
Rule: size is tiny or shape is square.
Rule: size is not enormous.
Rule: color is red.

Please propose a some new concepts, defined by a logical rule. ↙
    ↳ These new concepts can only refer to the following features:
- shape: triangle, rectangle, circle
- color: green, blue, yellow
- size: small, medium, large

Please make your rules short and simple, and please write your ↙
    ↳ response on a single line that begins with the text "Rule: ". ↙
    ↳ Provide 100 possible rules.
```

We generate from the above prompt at temperature=0, and split on line breaks to obtain candidate rules.

### A.4.2   Translating from natural language to Python

We translate Number Game concepts from English to Python via the following prompt for code-davinci-002, and generate at temperature=0 until linebreak:

```
# Write a python function to check if a number is $C$.
def check_number(num):
    return
```

We translate logical concepts from English to Python using a series of in-context examples, again generating with temperature=0 until the text #DONE is produced.[2]

```
def check_object(this_object, other_objects):
    """
    this_object: a tuple of (shape, color, size)
    other_objects: a list of tuples of (shape, color, size)

    returns: True if 'this_object' is positive according to the ↙
        ↳ following rule:
        Something is positive if it is not a small object, and not ↙
            ↳ a green object.
    """
    # shape: a string, either "circle", "rectangle", or "triangle"
    # color: a string, either "yellow", "green", or "blue"
    # size: an int, either 1 (small), 2 (medium), or 3 (large)
    this_shape, this_color, this_size = this_object

    # 'this_object' is not a part of 'other_objects'
    # to get all of the examples, you can use ↙
        ↳ 'all_example_objects', defined as 'other_objects + ↙
        ↳ [this_object]'
    # be careful as to whether you should be using ↙
        ↳ 'all_example_objects' or 'other_objects' in your code
    all_example_objects = other_objects + [this_object]

    # Something is positive if it is not a small object, and not a ↙
        ↳ green object.
    #START
    return (not this_size == 1) and (not this_color == "green")
#DONE

def check_object(this_object, other_objects):
    """
```

---

[2]This prompt is pretty long, and probably could be much shorter. Preliminary experiments suggested that a few in-context examples were very helpful, and so to increase the odds of the model working without much time spent prompt-engineering, we provided a large number of in-context examples.

```
    this_object: a tuple of (shape, color, size)
    other_objects: a list of tuples of (shape, color, size)

    returns: True if 'this_object' is positive according to the ↙
        ↳ following rule:
        Something is positive if it is bigger than every other object
    """
    # shape: a string, either "circle", "rectangle", or "triangle"
    # color: a string, either "yellow", "green", or "blue"
    # size: an int, either 1 (small), 2 (medium), or 3 (large)
    this_shape, this_color, this_size = this_object

    # 'this_object' is not a part of 'other_objects'
    # to get all of the examples, you can use ↙
        ↳ 'all_example_objects', defined as 'other_objects + ↙
        ↳ [this_object]'
    # be careful as to whether you should be using ↙
        ↳ 'all_example_objects' or 'other_objects' in your code
    all_example_objects = other_objects + [this_object]

    # Something is positive if it is bigger than every other object
    #START
    return all( this_size > other_object[2] for other_object in ↙
        ↳ other_objects )
#DONE

def check_object(this_object, other_objects):
    """
    this_object: a tuple of (shape, color, size)
    other_objects: a list of tuples of (shape, color, size)

    returns: True if 'this_object' is positive according to the ↙
        ↳ following rule:
        Something is positive if it is one of the largest
    """
    # shape: a string, either "circle", "rectangle", or "triangle"
    # color: a string, either "yellow", "green", or "blue"
    # size: an int, either 1 (small), 2 (medium), or 3 (large)
    this_shape, this_color, this_size = this_object

    # 'this_object' is not a part of 'other_objects'
    # to get all of the examples, you can use ↙
        ↳ 'all_example_objects', defined as 'other_objects + ↙
        ↳ [this_object]'
    # be careful as to whether you should be using ↙
        ↳ 'all_example_objects' or 'other_objects' in your code
    all_example_objects = other_objects + [this_object]

    # Something is positive if it is one of the largest
    #START
    return all( this_size >= other_object[2] for all_example_object ↙
        ↳ in all_example_objects )
#DONE

def check_object(this_object, other_objects):
    """
    this_object: a tuple of (shape, color, size)
    other_objects: a list of tuples of (shape, color, size)

    returns: True if 'this_object' is positive according to the ↙
        ↳ following rule:
        Something is positive if it is smaller than every yellow ↙
            ↳ object
    """
```

```python
    # shape: a string, either "circle", "rectangle", or "triangle"
    # color: a string, either "yellow", "green", or "blue"
    # size: an int, either 1 (small), 2 (medium), or 3 (large)
    this_shape, this_color, this_size = this_object

    # `this_object` is not a part of `other_objects`
    # to get all of the examples, you can use ↙
        ↘ `all_example_objects`, defined as `other_objects + ↙
        ↘ [this_object]`
    # be careful as to whether you should be using ↙
        ↘ `all_example_objects` or `other_objects` in your code
    all_example_objects = other_objects + [this_object]

    # Something is positive if it is smaller than every yellow object
    #START
    return all( this_size < other_object[2] for other_object in ↙
        ↘ other_objects if other_object[1] == "yellow" )
#DONE

def check_object(this_object, other_objects):
    """
    this_object: a tuple of (shape, color, size)
    other_objects: a list of tuples of (shape, color, size)

    returns: True if `this_object` is positive according to the ↙
        ↘ following rule:
        Something is positive if there is another object with the ↙
            ↘ same color
    """
    # shape: a string, either "circle", "rectangle", or "triangle"
    # color: a string, either "yellow", "green", or "blue"
    # size: an int, either 1 (small), 2 (medium), or 3 (large)
    this_shape, this_color, this_size = this_object

    # `this_object` is not a part of `other_objects`
    # to get all of the examples, you can use ↙
        ↘ `all_example_objects`, defined as `other_objects + ↙
        ↘ [this_object]`
    # be careful as to whether you should be using ↙
        ↘ `all_example_objects` or `other_objects` in your code
    all_example_objects = other_objects + [this_object]

    # Something is positive if there is another object with the ↙
        ↘ same color
    #START
    return any( this_color == other_object[1] for other_object in ↙
        ↘ other_objects )
#DONE

def check_object(this_object, other_objects):
    """
    this_object: a tuple of (shape, color, size)
    other_objects: a list of tuples of (shape, color, size)

    returns: True if `this_object` is positive according to the ↙
        ↘ following rule:
        Something is positive if it has a unique combination of ↙
            ↘ color and shape
    """
    # shape: a string, either "circle", "rectangle", or "triangle"
    # color: a string, either "yellow", "green", or "blue"
    # size: an int, either 1 (small), 2 (medium), or 3 (large)
    this_shape, this_color, this_size = this_object

    # `this_object` is not a part of `other_objects`
```

```
        # to get all of the examples , you can use ↙
            ↳ `all_example_objects`, defined as `other_objects + ↙
            ↳ [this_object]`
        # be careful as to whether you should be using ↙
            ↳ `all_example_objects` or `other_objects` in your code
        all_example_objects = other_objects + [this_object]

        # Something is positive if it has a unique combination of color ↙
            ↳ and shape
        #START
        return all( this_shape != other_object [0] or this_color != ↙
            ↳ other_object [1] for other_object in other_objects )
    #DONE

def check_object(this_object , other_objects):
    """
    this_object: a tuple of (shape , color , size)
    other_objects: a list of tuples of (shape , color , size)

    returns: True if `this_object` is positive according to the ↙
        ↳ following rule:
        Something is positive if it has the same color as the ↙
            ↳ majority of objects
    """
    # shape: a string , either "circle", "rectangle", or "triangle"
    # color: a string , either "yellow", "green", or "blue"
    # size: an int , either 1 (small), 2 (medium), or 3 (large)
    this_shape , this_color , this_size = this_object

    # `this_object` is not a part of `other_objects`
    # to get all of the examples , you can use ↙
        ↳ `all_example_objects`, defined as `other_objects + ↙
        ↳ [this_object]`
    # be careful as to whether you should be using ↙
        ↳ `all_example_objects` or `other_objects` in your code
    all_example_objects = other_objects + [this_object]

    # Something is positive if it has the same color as the ↙
        ↳ majority of objects
    #START
    majority_color = max(["yellow", "green", "blue"], key=lambda ↙
        ↳ color: sum(1 for obj in all_example_objects if obj [1] == ↙
        ↳ color))
    return this_color == majority_color
#DONE

def check_object(this_object , other_objects):
    """
    this_object: a tuple of (shape , color , size)
    other_objects: a list of tuples of (shape , color , size)

    returns: True if `this_object` is positive according to the ↙
        ↳ following rule:
        Something is positive if there are at least two other ↙
            ↳ objects with the same shape
    """
    # shape: a string , either "circle", "rectangle", or "triangle"
    # color: a string , either "yellow", "green", or "blue"
    # size: an int , either 1 (small), 2 (medium), or 3 (large)
    this_shape , this_color , this_size = this_object

    # `this_object` is not a part of `other_objects`
    # to get all of the examples , you can use ↙
        ↳ `all_example_objects`, defined as `other_objects + ↙
        ↳ [this_object]`
```

```
        # be careful as to whether you should be using ↵
            ↳ ‘all_example_objects‘ or ‘other_objects‘ in your code
        all_example_objects = other_objects + [this_object]

        # Something is positive if there are at least two other objects ↵
            ↳ with the same shape
        #START
        return sum(1 for other_object in other_objects if ↵
            ↳ other_object[0] == this_shape) >= 2
#DONE

def check_object(this_object, other_objects):
    """
    this_object: a tuple of (shape, color, size)
    other_objects: a list of tuples of (shape, color, size)

    returns: True if ‘this_object‘ is positive according to the ↵
        ↳ following rule:
        C
    """
    # shape: a string, either "circle", "rectangle", or "triangle"
    # color: a string, either "yellow", "green", or "blue"
    # size: an int, either 1 (small), 2 (medium), or 3 (large)
    this_shape, this_color, this_size = this_object

    # ‘this_object‘ is not a part of ‘other_objects‘
    # to get all of the examples, you can use ↵
        ↳ ‘all_example_objects‘, defined as ‘other_objects + ↵
        ↳ [this_object]‘
    # be careful as to whether you should be using ↵
        ↳ ‘all_example_objects‘ or ‘other_objects‘ in your code
    all_example_objects = other_objects + [this_object]

    # C
    #START
```

When modeling the concepts of *majority/minority color* we used a different prompt for converting natural language to python, because the above includes *majority color* as one of its in-context examples. We gave GPT-4 the following prompt for those concepts:[3]

```
Please write a python function to check if an object obeys a ↵
    ↳ logical rule. The logical rule talks about the following ↵
    ↳ features:

shape: a string, either "circle", "rectangle", or "triangle"
color: a string, either "yellow", "green", or "blue"
size: an int, either 1 (small), 2 (medium), or 3 (large)

The python function should be called ‘check_object‘, and inputs:

‘this_object‘: a tuple of (shape, color, size)
‘other_objects‘: a list of tuples of (shape, color, size)

The logical rule should check if ‘this_object‘ has a certain ↵
    ↳ relationship with ‘other_objects‘. Collectively, ↵
    ↳ ‘[this_object]+other_objects‘ correspond to all of the ↵
    ↳ objects, so if the rule references the whole examples, it is ↵
    ↳ talking about that structure.

The logical rule is: C
```

---

[3]We strongly suspect that GPT-4 with the following prompt is strictly better then Codex. The high cost of using GPT-4 makes it more practical to use the Codex prompt for the main experiments however.

```
Please start your response by writing the following code, and then ↙
    ↳ complete the function body so that it returns 'True' if and ↙
    ↳ only if the logical rule above holds.

```
def check_object(this_object, other_objects):
    """
    this_object: a tuple of (shape, color, size)
    other_objects: a list of tuples of (shape, color, size)

    returns: True if 'this_object' is positive according to the ↙
        ↳ following rule:
        C
    """
    # shape: a string, either "circle", "rectangle", or "triangle"
    # color: a string, either "yellow", "green", or "blue"
    # size: an int, either 1 (small), 2 (medium), or 3 (large)
    this_shape, this_color, this_size = this_object

    # 'this_object' is not a part of 'other_objects'
    # to get all of the examples, you can use ↙
        ↳ 'all_example_objects', defined as 'other_objects + ↙
        ↳ [this_object]'
    # be careful as to whether you should be using ↙
        ↳ 'all_example_objects' or 'other_objects' in your code
    all_example_objects = other_objects + [this_object]

    # return True if and only if:
    # C
```

## A.5 GPT-4 Baselines

Our GPT-4 baseline for each domain presented the examples $X_{1:K}$ in string form and then asked
GPT-4 to respond Yes/No as to whether a test example $X_{\text{test}}$ belonged to the same concept. GPT-4
was then queried at temperature=1 to collect 10 samples. Samples not beginning with 'y'/'n' were
discarded, and the ratio of remaining samples that began with 'y' was computed (case insensitive).

We show below example prompts for the number and logic domains.

```
Here are a few example number concepts:
-- The number is even
-- The number is between 30 and 45
-- The number is a power of 3
-- The number is less than 10

Here are some random examples of numbers belonging to a possibly ↙
    ↳ different number concept:
98, 81, 86, 93

Question: Does the number 42 belong to the same concept as the ↙
    ↳ above numbers?
Answer (one word, yes/no):
```

Logical concept example prompt:

```
Here are some example concepts defined by a logical rule:

Rule for Concept #1: Something is positive if it is the biggest ↙
    ↳ yellow object in the example
Rule for Concept #2: Something is positive if there is another ↙
    ↳ object with the same color in the example
Rule for Concept #3: Something is positive if it is the same color ↙
    ↳ as the smallest triangle in the example
```

```
Now please look at the following examples for a new logical rule.

    An Example of Concept #4:
        POSITIVES: none
        NEGATIVES: (large yellow circle), (small green circle), ↙
            ↳ (medium green circle), (small yellow triangle)
    Another Example of Concept #4:
        POSITIVES: (small green circle), (large green circle)
        NEGATIVES: (large yellow circle), (medium blue circle)
    Another Example of Concept #4:
        POSITIVES: (small green rectangle)
        NEGATIVES: (medium yellow circle), (medium blue rectangle), ↙
            ↳ (large green circle), (medium green circle)
    Another Example of Concept #4:
        POSITIVES: (medium green rectangle)
        NEGATIVES: (medium yellow circle), (small yellow ↙
            ↳ rectangle), (medium yellow rectangle), (medium blue ↙
            ↳ rectangle)
    Another Example of Concept #4:
        POSITIVES: (small green rectangle)
        NEGATIVES: (large yellow rectangle), (small yellow ↙
            ↳ triangle), (medium green circle), (small blue rectangle)
    Another Example of Concept #4:
        POSITIVES: (medium green triangle)
        NEGATIVES: (medium blue triangle), (medium blue rectangle), ↙
            ↳ (large blue triangle), (small yellow triangle)
    Another Example of Concept #4:
        POSITIVES: none
        NEGATIVES: (small yellow circle), (large blue circle)
    Another Example of Concept #4:
        POSITIVES: none
        NEGATIVES: (large green circle), (small blue rectangle), ↙
            ↳ (small green triangle), (medium blue rectangle)
    Another Example of Concept #4:
        POSITIVES: (small green rectangle)
        NEGATIVES: (small yellow circle), (large blue rectangle)

Now we get a new collection of examples for Concept #4:
(medium blue triangle) (large yellow triangle) (small blue ↙
    ↳ rectangle) (large blue circle) (small yellow circle)
Question: Based on the above example, is a (small yellow circle) in ↙
    ↳ the concept?
Answer (one word, just write yes/no):
```

## A.6 Latent Language Baseline

For fair comparison, we designed our latent language baseline to be as similar to our system as possible. It performs maximum likelihood estimation of a single concept, rather than estimate a full posterior, but uses the exact same prompts and likelihood functions as our model. The most important difference from the original latent language paper [31] is that instead of training our own neural models for language interpretation and language generation, we use pretrained models (Codex/code-davinci-002 and GPT-4).

## A.7 Ablation of the proposal distribution

We ablate the proposal distribution by proposing hypotheses unconditioned on $X_{1:K}$. We accomplish this by drawing concepts from the following alternative prompt, which is designed to resemble the prompt used by the full model except that it does not include $X_{1:K}$:

```
# Python 3
# Here are a few example number concepts:
# -- The number is even
# -- The number is between 30 and 45
```

```
# -- The number is a power of 3
# -- The number is less than 10
# -- The number is
```

## A.8 Pretrained prior

Our pretrained prior comes from the opensource model CodeGen [44], which was trained on source code. We chose this model because we suspected that pretraining on source code would give better density estimation for text describing precise rules. We formatted the rules as a natural language comment and prefixed it with a small amount of domain-specific text in order to prime the model to put probability mass on rules that correctly talk about numbers or shapes.

For the number game, we would query CodeGen for the probability of $p(C)$ via

```
# Here is an example number concept:
# The number is C
```

For the number game's code baseline, we would query CodeGen for the probability of $p(C)$ via

```
# Python 3
# Let's think of a number concept.
# Write a python function that returns true if `num` belongs to ↵
    ↳ this number concept.
def check_if_in_concept(num):
    return C
```

For logical concepts we would query CodeGen for the probability of $p(C)$ via

```
# Here are some simple example shape concepts:
# 1. neither a triangle nor a green rectangle
# 2. not blue and large.
# 3. if it is large, then it must be yellow.
# 4. small and blue
# 5. either big or green.
# 6. C
```

Because the proposal distribution would generate rules beginning with the prefix "Something is positive if..." we would remove that text before computing $p(C)$ as above.

