# A   Appendix

## A.1   Supplemental Results

Fig. 6 illustrates model predictions across every Number Game concept in [33].

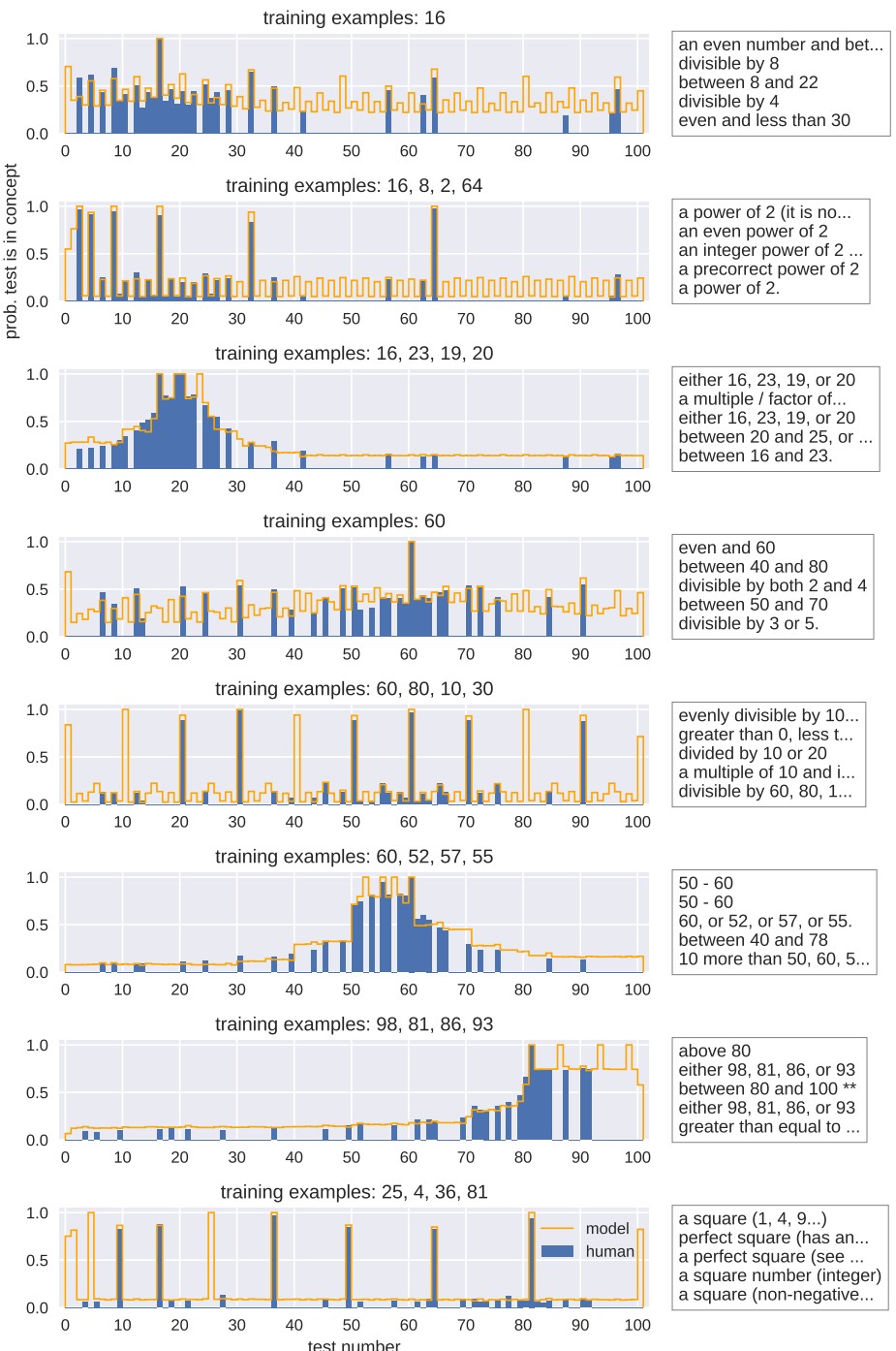

Figure 6: Model predictions across every Number Game concept in [33]

Recall that we deduplicated the proposals instead of performing actual importance sampling. Fig. 7 contrasts model fit for importance sampling and deduplication. We originally did deduplication simply because importance sampling is not possible with GPT-4, and GPT-4 proved necessary for the logical concepts. On number concepts we used code-davinci-002, from which we can construct an importance sampler because it exposes the log probability of its samples. On number concepts deduplication provides a fit that is on-par (actually slightly better) compared to importance sampling (Fig. 7).

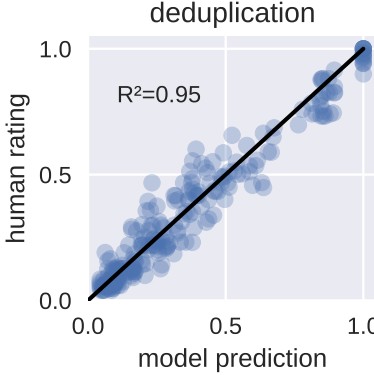 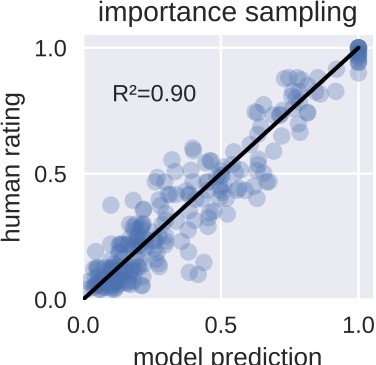

Figure 7: Monte Carlo inference using deduplication instead of importance sampling does not harm model fit to human data. The above figures show Number Game models using a learned prior and 100 samples, and show predictions only on holdout data.

## A.2   Human Study

16 participants were recruited primarily through a Slack message sent to a channel populated by members of our academic department. Participants had an average age 28.1 (stddev 13.6, all over 18), and were 7 male/5 female/1 nonbinary/3 declined to answer. Participants were randomly split between the concepts of *most common color* / *least common color*. Each participant went through 15 trials, and took an average of 294s to complete those 15 trials. In exchange for participating in the study, participants received $10 in Amazon gift cards. Fig. 8 illustrates the web interface shown to our human participants, including the cover story.

## A.3   Modeling

### A.3.1   Temperature and Platt Transform

Adding a temperature parameter $T$ to a model corresponds to computing the posterior via

$$p_{\text{Temp}}(X_{\text{test}} \in C | X_{1:K}) \approx \sum_{C \in \{C^{(1)},...,C^{(S)}\}} w^{(C)} \mathbb{1}\left[X_{\text{test}} \in C\right], \text{ where}$$

$$w^{(C)} = \frac{\left(\tilde{w}^{(C)}\right)^{1/T}}{\sum_{C'} \left(\tilde{w}^{(C')}\right)^{1/T}} \text{ and } \tilde{w}^{(C)} = p(C)p(X_{1:K}|C)\mathbb{1}\left[C \in \{C^{(1)}, \ldots, C^{(S)}\}\right]$$

$$(10)$$

Adjusting the predictions of a model using a Platt transform corresponds to introducing parameters $a$ and $b$ which transform the predictions according to

$$p_{\text{Platt}}(X_{\text{test}} \in C | X_{1:K}) = \text{Logistic}(b + a \times \text{Logistic}^{-1}\left(p(X_{\text{test}} \in C | X_{1:K})\right)) \quad (11)$$

For the number game, every model has its outputs transformed by a learned Platt transform. This is because we are modeling human ratings instead of human responses. We expect that the ratings correspond to some monotonic transformation of the human's subjective probability estimates, and so this transformation gives some extra flexibility by inferring the correspondence between probabilities and ratings. Logical concept models do not use Platt transforms.

# Trial 1: Please read these instructions carefully

You are going to attempt to learn the meaning of a new word in an alien language, which the aliens call "Wudsy." On each trial, you are going to see a collection of shapes at the bottom of the webpage, and your job is to select which ones you think are "Wudsy." Afterward, the aliens tell you which shapes are "Wudsy."

The meaning of the word Wudsy is the same during the whole experiment. However, it is possible that whether something is Wudsy depends on what other shapes it is in the context of. Wudsy may or may not correspond to an English word.

To start with, no one has given you any examples of what counts as "Wudsy." So just do your best below and pick which ones you think might belong to the concept called "Wudsy." Right after you do so, the aliens are going to label the Wudsy objects by drawing a black box around them, and then you are going to get another round of guessing which objects are "Wudsy."

You will go through 15 trials of guessing what counts as Wudsy. Remember that the meaning of Wudsy does not change during the experiment, but it might depend on the other shapes in the collection.

(trial 1/15) Click yes on the objects that you think are Wudsy, and No on the other objects. Then click Next.

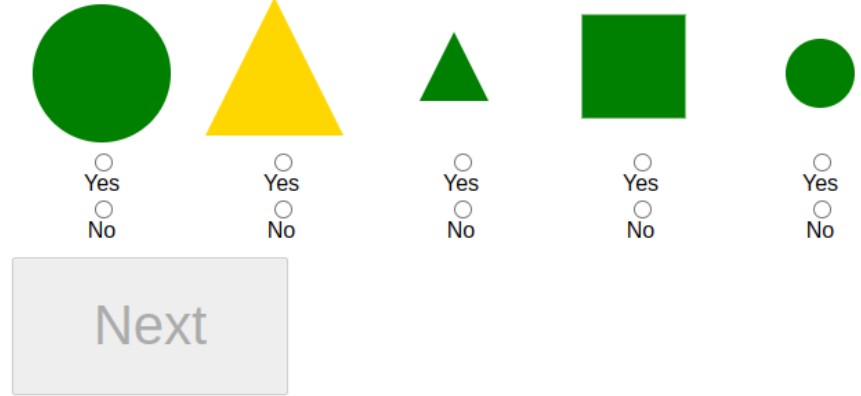

Figure 8: Cover story and web interface for our version of the logical concept learning study, which is based on [45]

### A.3.2 Parameter fitting

Training consists of fitting the parameters $T$, $\theta$ (for the prior), $\epsilon$ (for the likelihood), $\alpha$ and $\beta$ (for the logical concepts likelihood), and Platt transform parameters $a$, $b$ (for the Number Game). In practice, this amounts to around 400 parameters, almost all of which come from $\theta$.

We fit these parameters using Adam with a learning rate of 0.001. We perform 1000 epochs of training for the Number Game, and 100 epochs for logical concepts. There is a tenfold difference in the number of concepts, so this way they take about the same number of gradient steps.

For the number game we do 10-fold cross validation to calculate holdout predictions. For logical concepts we use the train-test split introduced in [45], which involves running different groups of human subjects on each concept twice, with different random examples. One sequence of random examples is arbitrarily designated as training data, and the other as holdout data.

All model were trained on a laptop using no GPUs. Training takes between a few minutes and an hour, depending on the domain and the model.

Some of the parameters that we fit, namely $\epsilon$, $\alpha$, $\beta$, cannot be negative. To enforce this we actually optimize the inverse logistic of those parameters.

### A.3.3 MCMC over Logical Expressions

Fleet was used[1] to perform MCMC over logical expressions with the domain-specific primitives in this file, which include:

$$\text{true}, \text{false} : \text{boolean}$$
$$\text{blue}, \text{yellow}, \text{green} : \text{object} \rightarrow \text{boolean}$$
$$\text{rectangle}, \text{circle}, \text{triangle} : \text{object} \rightarrow \text{boolean}$$
$$\text{small}, \text{medium}, \text{large} : \text{object} \rightarrow \text{boolean}$$