# OpenReview forum: "Human-like Few-Shot Learning via Bayesian Reasoning over Natural Language"
_NeurIPS.cc/2023/Conference — NeurIPS 2023 oral_

### Official Review · Reviewer_8GEn · 2023-06-29

**Soundness:** 4 excellent
**Presentation:** 3 good
**Contribution:** 4 excellent
**Rating:** 7
**Confidence:** 4

**Summary:**

This paper presents a model for how humans learn abstract symbolic concepts from induction. The model uses an off-the-shelf language model as a meta-prior, which is then tuned to form a task-specific prior over hypotheses using a small number of human samples. This prior is then incorporated into a Bayesian inference setup to solve inductive reasoning tasks. The model closely adheres to human judgments and also seems to show that natural language is a better performing hypothesis space than programs.

**Strengths:**

- Clear and concise
- Thoughtful discussion
- Novel incorporation of LLMs into cognitive modeling

Comments:
- Line 162: Super cool result. I’d love if you could stay on this result a little longer and speculate why this might be.
- Line 275: You’re basically using LLMs as a meta-prior, and then tuning it to obtain a task-specific prior. This is very interesting.

**Weaknesses:**

- The abstract is vague. I’d recommend the authors expand the abstract in length and make reference to their results.
- Some more implementation details in Sections 4 and 5 would be helpful for future readers. Please see questions below.

**Questions:**

Major:
- Do you have thoughts on exactly how "strong" you should make the prior and likelihood functions? For example, I'd imagine you might get even better raw performance by using GPT-3 as the prior model. Is there a sweet spot wrt model capabilities for matching human judgments? If so, what are the implications for your rational process model?

Minor:
- Line 36: ”regularizes the learner toward probable generalizations” Some references here would be nice—you can just duplicate the ones you have later in the text.
- Line 48: Why is this well-suited for natural language?
- Line 83: typo “has also”
- Line 135: What’s the sampling technique—nucleus sampling, greedy, etc? I’d like some more detail on the implementation of the proposal distribution, because sampling and sampling technique can change the effective distribution a lot.
- Line 142: Can you just explain how you fit the parameters right after you introduce them? I was wondering how you trained the parameters for a few paragraphs.

Nits:
- Figure 3: Do you have a higher resolution screenshot?

**Limitations:**

Yes, the authors have addressed limitations.

---

> ### Author Rebuttal · Authors · 2023-08-02
>
> Thank you for the thoughtful review and kind words. Below we respond to some of your main points, but can discuss further during the discussion period:
>
> > Do you have thoughts on exactly how "strong" you should make the prior and likelihood functions?
>
> Our thoughts are that the proposal distribution $q$ needs to be the "strongest", because it needs to propose plausible hypotheses from scratch. In our paper, the likelihood $p(X|C)$ only needs to translate English to Python, so it requires a decent LLM or a fine-tuned smaller model (we likely could have fine-tuned a T5-like model also).
>
> In the global response, we have attached a PDF showing analysis of using smaller open-source models for the proposal distribution and the likelihood. These new results suggest, as above, that $q$ needs to be the "strongest", but that a weaker $q$ can still work, provided that you take more samples. This introduces an interesting tradeoff between the "strength" of $q$, and the amount of compute (# samples) that are taken at test time.
>
> The prior $p(C)$ is only responsible for giving a soft bias toward simpler, shorter language, and in our experience, does not require a "strong" model: we tried both a 350M open source model and tuning a ~100M model.
>
> > Is there a sweet spot wrt model capabilities for matching human judgments? If so, what are the implications for your rational process model?
>
> That is an interesting scientific question. For example, a stronger proposal distribution might sometimes outperform humans, which could suggest certain limits on bottom-up psychological processes. Although our paper does not specifically explore the questions you raise, they would make for thrilling future work. We will add this to the discussion.
>
> > What’s the sampling technique—nucleus sampling, greedy, etc?
>
> The proposal distribution was sampled with temperature $T=1$ and $\text{topP}=1.0$., which effectively disables nucleus sampling. Because we are taking multiple proposals, a higher temperature made more sense to encourage diversity.
>
> For the likelihood, which calls out to an LLM to translate English to Python, we sampled with temperature $T=0$, both because Codex was very reliable at this translation, and to avoid needless stochasticity.
>
> The revision will mention these issues. Thanks for the catch.
>
> > Why is this well-suited for natural language? [prime numbers less than 30]
>
> Perhaps a better phrasing would be that these concepts are "suitable to be expressed in natural language". All that is meant is that it is possible and reasonably practical to express these concepts in words.
>
>
>
> > Can you just explain how you fit the parameters right after you introduce them? I was wondering how you trained the parameters for a few paragraphs.
>
> Yes, we can move up the explanation of parameter fitting in Section 4.
>
> > Figure 3: Do you have a higher resolution screenshot?
>
> No we don't: this image was provided to us by Piantadosi et al. 2016 (with permission).

---

> > ### Comment · Reviewer_8GEn · 2023-08-14
> >
> > Thanks for answering my questions. The hyperparameter choices for sampling make sense to me, and my concerns have been adequately addressed. I have increased my score to a 7.
> >
> > I still think this paper would be better served by a longer abstract. For example, calibration is at the core of this paper, and this is elided into "can be fit to human data" in the abstract. Unpacking and explaining calibration with an extra sentence would make sense to me. However, if the authors feel committed to the short abstract, I will not continue belaboring this point and leave it to their discretion.

---

> > > ### Author Response · Authors · 2023-08-14
> > >
> > > Thanks for the response, and for the increase to your score.
> > >
> > > > I still think this paper would be better served by a longer abstract
> > >
> > > Agreed: and if accepted, we'll have an extra page to use for expanding the abstract and providing further details and discussion throughout the paper.

---

### Official Review · Reviewer_qc38 · 2023-07-02

**Soundness:** 3 good
**Presentation:** 4 excellent
**Contribution:** 4 excellent
**Rating:** 8
**Confidence:** 3

**Summary:**

This paper proposes a model for human learning of concepts from few examples (aka "few shot" setting) that leverages natural language as an internal concept representation. This is a key trick because it means that

* an LLM can be used to act as a proposal distribution for (efficiently) generating data-dependent candidate concepts
* a prior distribution over concept space can be tuned (estimated) from human judgments using NLP features
* a "likelihood function" for determining whether a concept could have generated an observation can be implemented via a "text2code" LLM that derives Python code from the natural language concept representation

Experiments on "Number Game" and logical concept learning show concept learning with "psychologically plausible" sample complexity, remarkable agreement with human judgments, and explainable failure modes. In comparison with state-of-the-art Bayesian Program Learning (BPL) this approach is able to search through a much smaller hypothesis space (thanks to "high quality" natural language candidate proposals) but also generalize to totally novel concepts which are not express-able in the BPL due to the flexibility of natural language vs the primitives available to the BPL approach.

**Strengths:**

The paper is well-contextualized with respect to related work. It identifies a major open question (tractability vs expressivity) and addresses it via a key insight (using natural language is now much more feasible with recent LLM advances). Actually implementing this idea requires multiple non-obvious steps to bring it all together. The experiments systematically illuminate both the power and limitations of the approach in controlled but challenging problem settings. The agreement with human subject results gives the approach credibility. The comparison approach (Bayesian Program Learning) is a strong baseline. Overall I found this work to be a creative and well-executed integration of LLM advances into Bayesian cognitive modeling.

**Weaknesses:**

I guess if we were purely looking for some kind of "predictive performance" task there might be other approaches that "outperform" the framework here. However, the grounding in cognitive science for this work means that is probably not the right criterion, but rather this work is pursuing improved understanding of learning itself.

I'd be interested to see a discussion or mention of some task or problem setting that successfully "red teams" this approach - ie, some problem which is adversarially chosen to be challenging or ill-suited for this framework (while still remaining in-scope within the domain of non-embodied abstract concept learning).

The approach seems critically dependent on the quality of the Python code generation aspect. From the supplemental materials it looks like a lot of prompt engineering went into getting that to work. To some extent this seems like kind of a "weak link" for the whole enterprise: using natural language as the concept space requires the availability of a (very general) mechanism for computing observation likelihood or concept membership given the concept.


**Questions:**

L92: it's a little unclear what it means to say that P(X_{test} \in C) - it doesn't seem exactly to be P(X_{test} | C), or even P(X_{test}, C), but rather more like the posterior probability that X_{test} was generated by _the same latent concept C that generated X_{1:K}_. I guess the key assumption here is that \mathbb{1}[X_{test} \in C] is easily computable, but that isn't clear or established by this point in the development (later it is shown how to do this with NL-to-Python).

L147: this summary of all the variations is pretty dense and takes a bit of reader effort to unpack.

Section 4: Is there a stronger or more recent baseline for Number Game, vs Latent Language from 2018?

**Limitations:**

Limitations seem reasonable and likely avenues for future work.

The reproducibility issues with using GPT4 seem addressed by including the responses in the software/data release.

---

> ### Author Rebuttal · Authors · 2023-08-02
>
> Thank you for the thoughtful review and enthusiastic support. Please find below our clarifications:
>
> > …mention of some task or problem setting that successfully "red teams" this approach - ie, some problem which is adversarially chosen to be challenging or ill-suited for this framework (while still remaining in-scope within the domain of non-embodied abstract concept learning).
>
> Because we use a large language model as a proposal distribution, problems which LLMs struggle with would likely foil our approach. However, there is one caveat: because we draw multiple proposals (~100), and because we reweigh each of them using Bayes Rule, the model only needs a few proposals that hit the mark. Therefore, our approach could be “red teamed” by problems that cause LLMs to “fall on their face,” but probably not by problems that cause LLMs to become merely “flakey”.
>
> Also, our current approach translates each natural language concept into Python, so it would struggle to learn concepts that are not easily expressible in a precise formal language like Python. This restriction is not strictly necessary: you could instead define the likelihood $p(X|C)$ using another neural model in order to allow learning "fuzzier" concepts, although this would probably come at the expense of precision. The last sentence of the paper alludes to these subtleties by saying "We bypass natural language’s ambiguity by translating it into Python for the likelihood computation, but future work needs to determine if language models can produce language precise enough for induction, or if refining into languages like Python is more practical." If accepted, we'd have an extra page for expanding that point with this discussion.
>
> > …critically dependent on the quality of the Python code generation aspect. From the supplemental materials it looks like a lot of prompt engineering went into getting that to work
>
> For the initial submission, we tried exactly two prompts for converting logical concepts into python: a short simple prompt (which was unreliable), and later a very long prompt, which proved reliable (but probably went overboard). In the weeks after the submission, we also tried prompting GPT-4 with instructions but no few-shot examples, which works better than the long Codex prompt used in the initial submission.
>
> In our experience, though, converting simple natural language utterances into snippets of python requires little prompt-engineering, provided one is willing to use larger models such as Codex. In the global response, we also describe new results using Llama-2 70B, a very recent open source LLM, to convert natural language into Python, which required zero new prompt engineering.
>
> > I guess if we were purely looking for some kind of "predictive performance" task there might be other approaches that "outperform" the framework here. However, the grounding in cognitive science for this work means that is probably not the right criterion, but rather this work is pursuing improved understanding of learning itself.
>
> Indeed, our primary aims are scientific. After the submission deadline, we tried optimizing the logical concept model to maximize average task performance, instead of maximizing fit to humans. We found that the maximizing average performance makes the model surpass human performance, but also degrades human-model agreement. If accepted, we will include these results in the revision.
>
> > Section 4: Is there a stronger or more recent baseline for Number Game, vs Latent Language from 2018?
>
> Thanks for the suggestion. We have now ran a DreamCoder baseline (a recent neurosymbolic Bayesian program learner). It achieves a decent (but not great) fit to the human Number Game data: $R^2=.75$, which should be contrasted width $R^2=.95$ for our full model.
>
> > it's a little unclear what it means to say that $P(X_\text{test} \in C)$ ...  I guess the key assumption here is that $\mathbb{1}[X_\text{test} \in C]$ is easily computable
>
> We like how you put it: The key assumption is that $\mathbb{1}[X_\text{test} \in C]$ is easily computed, which comes from translating $C$ into Python. A footnote will be added clarifying this.

---

> > ### Comment · Reviewer_qc38 · 2023-08-15
> > **Thanks for the response**
> >
> > Thanks for the response elaborating and clarifying with respect to my questions. The additional results described (maximized predictive perf and DreamCoder baseline) and details about different prompt/model variations tried would strengthen the submission even further, which would be great to see.

---

### Official Review · Reviewer_ce9a · 2023-07-07

**Soundness:** 4 excellent
**Presentation:** 4 excellent
**Contribution:** 4 excellent
**Rating:** 10
**Confidence:** 4

**Summary:**

The authors introduce a computational approach for explaining how humans perform few-shot concept learning. The proposal is that humans use Bayesian inference over natural-language definitions of concepts. In more detail, a bottom-up proposer generates a set of candidate concept definitions, and Bayesian inference is then performed over these candidates - based on a prior distribution over concept definitions and a likelihood capturing how well the definition covers the observed examples.

The paper then presents two main experiments that implement the proposal via the use of recent large language models (LLMs). The prior over natural-language concept definitions is the probability that an LLM assigns to the definition. The proposer is instantiated as the responses of an LLM prompted with the training examples. The likelihood is computed by translating the proposed concept definitions into Python code using a code-based LLM and then running the Python code on the training examples. In both case studies (the Number Game and learning compositional logical concepts), the proposed approach provides a strong fit to human data.


**Strengths:**

- S1: The approach provides an interesting combination of Bayesian inference and neural network models that combines complementary strengths of both approaches: the powerful inference abilities of Bayesian models and the tractability and flexibility of neural models. This combination results in a system that is arguably more powerful than either type of approach on its own, representing an important advance in computational cognitive science.
- S2: The components of the approach are well-motivated by high-level considerations and are well-operationalized using current AI tools.
- S3: The results are compelling: in both case studies, the proposed model shows advantages (in overall fit to human data and/or in computational tractability) over strong baselines on both the Bayesian side and the neural network side.
- S4: Working within this proposed paradigm, the authors show how to fit a model to human data in a way that successfully transfers human priors into a neural model.
- S5: This work will likely be useful for future researchers working in Bayesian modeling and/or neural networks as a way to take insights from one school of thought and use them to overcome weaknesses of the other school of thought.


**Weaknesses:**

- W1: Natural language is ambiguous, so natural language strings do not really provide clear concept definitions - and, by extension, inference over natural language strings cannot strictly be viewed as inference over concept definitions. (I.e., in the general case, natural language cannot be unambiguously translated into Python code). That said, as the experiments show, it is clearly close enough to work very well in at least some settings. In addition, the authors are clear in stating (in lines 286 to 293) that they do not claim that natural language is the language of thought but rather that it is a useful heuristic tool for modeling human thinking.
- W2: The specific proposal is well-suited to concepts that can be naturally expressed in language, but is not suited to concepts that do not have a straightforward linguistic definition. The authors acknowledge this point (lines 48 to 51).
- W3: It was a little unclear to me what the paper’s main contribution is: Is it (i) a new hypothesis about how humans perform few-shot learning, or is it (ii) a way to make tractable some previously-existing hypotheses that were previously intractable to evaluate? Both types of contribution are valuable, but it would be helpful to clarify which is/are being made here. It’s clear that the paper accomplishes (ii), but it is not obvious to me that it does (i), as the basic high-level ideas seem to be present in the cited prior work (e.g., ideas about performing Bayesian inference over a small-ish number of heuristic proposals are present in prior work about bounded rationality). It’s not a problem if (i) is not done here, but if it is done, it would be helpful to clearly state what new hypothesis is being offered, and if it is not done, it would be helpful to state explicitly that (ii) is the main type of contribution being made. One reason I’m confused here is due to differing senses of the word “model”: the paper clearly states that it proposes a new model, but I’m not sure if this is meant in the sense where “model” means “hypothesis” or the sense where it means something more like “implementation”.


**Questions:**

It would be helpful to hear your thoughts on the point discussed in W3 above, under “weaknesses.”

**Limitations:**

The authors do a strong job of discussing and acknowledging limitations

---

> ### Author Rebuttal · Authors · 2023-08-06
>
> Thank you for the thoughtful input and enthusiastic support. We are happy to correspond more during the discussion period, but here we mainly just address the specific question you raised:
>
> > …unclear to me what the paper’s main contribution is: Is it (i) a new hypothesis about how humans perform few-shot learning, or is it (ii) a way to make tractable some previously-existing hypotheses that were previously intractable to evaluate? … It’s clear that the paper accomplishes (ii), but it is not obvious to me that it does (i)
>
> Thank you for this helpful way of thinking about the different contributions of the work.
>
> Our main contribution is (ii), offering a computational model that makes inference tractable for certain very expressive hypothesis spaces that have proved valuable in cognitive modeling (Line 31: "Our goal is to build a model of humanlike concept learning that makes progress toward resolving the tension between intractable inference and expressive hypothesis classes"). There is also a more speculative account of our contributions (i), namely that our model offers a hypothesis for how human brains resolve the “curse of a compositional mind” (Spelke 2022: freeform recombination of concepts yields a combinatorial explosion). In this more speculative hypothesis, culturally-transmitted concepts and linguistic schemas help guide inner thought, making combinatorial thinking more tractable. We've shied away from making that claim, because it is not directly supported by our findings, though it is not contradicted by our work, either. Right now the manuscript tries to strike a balance by simply saying that "Natural language, even if it is not actually the same as our inner mental language, acts as a vast reservoir of human concepts, and provides a flexible algebra for combining them. Thus our best near-term strategy for modeling human thinking may be  to use natural language as a *heuristic approximation* to an inner Language of Thought" (emphasis added), later refining the claim from natural language generally to large language models specifically as a "reasonable surrogate for this [human] bottom-up [proposal] process, even if it its inner workings might differ greatly from human bottom-up proposal processes".
>
> We hope that the paper's current wording manages to strike the right balance, and are happy to revise in order to more clearly signpost the actual contributions, results, and concrete hypotheses, especially given that, if accepted, we have an additional page to include discussion and add clarifications.
>
> Last, about the other weaknesses you raise:
> > [the work is] not suited to concepts that do not have a straightforward linguistic definition... the authors are clear in stating (in lines 286 to 293) that they do not claim that natural language is the language of thought but rather that it is a useful heuristic tool for modeling human thinking.
>
> > Natural language is ambiguous, so natural language strings do not really provide clear concept definitions - and, by extension, inference over natural language strings cannot strictly be viewed as inference over concept definitions. The authors acknowledge this point
>
> Indeed, we *do not* provide a unified theory of human few-shot concept learning. Thank you for pointing out that the submission is careful about clearly discussing the limits of the work.

---

> > ### Comment · Reviewer_ce9a · 2023-08-19
> > **Thanks for the reply!**
> >
> > Thank you for the reply, which is very helpful for clarifying the few things I found unclear about the paper. I continue to view this paper highly and to enthusiastically recommend acceptance.

---

### Official Review · Reviewer_y5Ba · 2023-07-11

**Soundness:** 3 good
**Presentation:** 3 good
**Contribution:** 3 good
**Rating:** 7
**Confidence:** 4

**Summary:**

The paper proposes an approach to scaling up intractable Bayesian models of few-shot concept learning. The key idea is to (1) train an amortized posterior distribution q(C|X_{1:K}) over concepts (represented as natural-language expressions) and then (2) make predictions about membership by marginalizing over a finite set of latent concepts sampled from q. Several variants of this approach are considered, all of which use a large language model (Codex) for the likelihood function p(X | C) and proposal distribution q. The approach is evaluated on two few-shot learning experiments: a generative number concept task, and a discriminative logical concept task. A version of the model that reweights sampled concepts according to a prior distribution trained on human judgements is found to fit human patterns better than alternative models which either (1) replace the human-tuned prior scores with an off-the-shelf language model (CodeGen), (2) replace the natural-language prior with a Python prior, or (3) entirely omitting the separate proposal distribution q and sampling proposed concepts directly from the (learned) prior instead.

**Strengths:**

The manuscript presents a promising and potentially influential approach to scale up Bayesian models on few-shot learning tasks. A number of timely ideas are explored, and the results compare favorably with other recent approaches that have been well-received at NeurIPS (e.g. Bayesian program learning, and other neurosymbolic approaches). Many of the basic ideas in play are pretty familiar by now (e.g. amortizing a proposal distribution to avoid costly search over hypotheses, using natural language as a more expressive latent hypothesis space, fine-tuning on human priors to impart human-like inductive biases). But this work still 'remixes' and integrates them in an interesting way: for example, other language-guided BPL approaches (e.g. Wong et al, 2021, ICML; Andreas et al., 2017) have required a set of language annotations to fine-tune on, rather than utilizing more generic priors (although Codex would be more impractical for multi-modal tasks, conditioning on images). Evaluating on human behavior (rather than synthetic benchmarks) are another strength.

**Weaknesses:**

My primary concerns are around how to do credit assignment to the many potentially brittle component parts of the pipeline, and the applicability of this scaling approach outside of toy domains. One somewhat deflationary critique is that there is no Bayesian *inference* at all in this pipeline, and certainly no inference *over natural language*. The approach depends on the independent existence of a sufficiently powerful amortized posterior distribution from a very expensive inference procedure that has already been performed off-stage (i.e. training Codex). the model comparisons simply show more or less efficient ways to approximate a *predictive* distribution by marginalizing over high-probability regions of that independently pre-existing posterior. This isn't inference! Arguably, it's just wrangling the amortized product of an earlier inference, exploring different ways of leveraging human data to correct distortions in this mammoth amortized posterior and project it down to specific tasks at hand.

I still think this kind of 'wrangling' work is interesting and novel, as there are clearly more or less effective ways to do it, but (1) I believe the framing of actually doing Bayesian inference over natural language is misleading (including statements like 'our work adds Bayesian inference'), and (2) I would suggest much more focus and scrutiny on the black-box, closed-source models that are the 'wizard behind the curtain,' the parts of the pipeline actually responsible for the few-shot concept induction. For example, it is stated that Codex was used 'because we hypothesized that training on source code would transfer to reasoning about numbers,' but no other choice was considered.

I would like to see a bigger space of candidate posteriors compared. I would also like to see a stronger 'credit assignment' analysis testing components of the Codex pipeline (which I understand has now been deprecated, making these results difficult to reproduce?) For example, one very strong assumption is that each linguistically expressed concept maps deterministically to a single Python program, which in turn, can be evaluated deterministically on each number. There may be multiple valid programs corresponding to each linguistic utterance. What if the linguistic concept is good but the translation to python is bad? i.e. what if the best cutting-edge model were used as the proposal distribution, but a weaker code-generation model were used in the likelihood to evaluate the generated concepts on numbers? The current analysis cannot pull these apart.

**Questions:**

1. It wasn't clear how train-test splits were handled. The Fig. 2 caption alludes to 'held-out' judgements, but what was the split? How many examples were used to tune the prior? Were entire concepts held out, or were some human data seen for each example set?

2. In Fig. 2, it looked like the 'no proposal dist' model might potentially be competitive if it was given more than 100 samples to hit on good descriptions. This is still a relatively small number of samples, and even if it is somewhat expensive, it would aid understanding to extend the x-axis one more degree of magnitude.

3. It looked like data came from the extremely small, cherry-picked set of concepts used in the original Tenenbaum dataset (with N=8). However, much larger and more systematic datasets are now standard, e.g. Bigalow & Piantadosi, https://doi.org/10.5334/jopd.19, releasing 272k judgements using many more concepts. I would strongly suggest reporting generalization performance to these new concepts.

4. It wasn't clear how the Latent Language model actually worked for these task; it would help to clarify precisely how it differed from the other models in this case (did it just take the single maximum likelihood concept from the proposal distribution instead of marginalizing proportional to each sample?)

5. Did the likelihood function assume boolean output from the codex-derived python code? how was this ensured? what happened if the generated python code returned an error or non-boolean?

6. Section 5 states: "Except we now have a discriminative learning problem instead of a generative one" -- except wasn't the number game treated as discriminative by the model, using an indicator function (effectively deriving a disciminative classifier for whether each number is in or out the concept?)

7. I would have liked to see a more reasonable neurosymbolic BPL baseline, which actually does proper Bayesian inference (i.e. MCMC) using the latent concept space as the proposal distribution over valid programs.

A related more recent paper that may be worth including:

* Wong, L., Grand, G., Lew, A. K., Goodman, N. D., Mansinghka, V. K., Andreas, J., & Tenenbaum, J. B. (2023). From Word Models to World Models: Translating from Natural Language to the Probabilistic Language of Thought. arXiv preprint arXiv:2306.12672.

**Limitations:**

A clear limitation is that a central pillar of the approach, the Codex API, is now deprecated and unavailable to other researchers. It is therefore not clear whether any of the results can be replicated, creating further incentive for the authors to compare performance on a number of other amortized posteriors (proposal functions) and likelihood functions, ideally those which are open and maintained.

---

> ### Author Rebuttal · Authors · 2023-08-02
>
> Thank you for the thoughtful input, kind words, and suggested improvements. We address the main issues, with **new results bolded**, but can answer all of your questions during the discussion period.
>
> > My primary concerns are around how to do credit assignment to the many potentially brittle component parts of the pipeline
>
> In our view, the key ideas are (1) natural language as a hypothesis space, (2) Bayesian reasoning for forming predictions, and (3) LLMs for tractable inference. Achieving the full suite of results requires all ingredients, established by comparing against Bayesian Program Learning (no natural language), latent language (no Bayes), latent source code (LLMs, but no natural language), and no proposal dist (no bottom-up proposals for tractability).
>
> Our pipeline has 3 components: prior; likelihood; and proposal distribution. We ablate the prior by not tuning it and by removing it (in latent language). The proposal distribution was also ablated. The likelihood is so critical for discarding erroneous hypotheses that we did not even consider a model without it, but **we've now run a likelihood ablation (global response PDF), confirming that the likelihood computation is essential.**
>
> > what if the best cutting-edge model were used as the proposal distribution, but a weaker code-generation model were used in the likelihood
>
> To understand what happens when the individual components are merely made weaker instead of ablated entirely, we've run **new experiments on Llama-2, a recent 70B open-source model that is thought to be weaker than Codex (see attached global response PDF).** The new data show that a weaker LLM, like Llama-2, can implement effective likelihood models (act as code-generators), but stronger LLMs are important for proposal distributions, especially in the low-sample regime. Although these Llama-2 results do not change our scientific conclusions, they help in understanding how to practically engineer systems like ours using off-the-shelf tools.
>
> > Codex... has now been deprecated, making these results difficult to reproduce
>
> Codex is deprecated but freely available for academic use (with a special application needed), or on Azure (expensive, but for anyone). Ultimately, any closed-source model hurts reproducibility, and will be eventually deprecated. To mitigate this, we've archived our OpenAI queries+responses, as noted in the author checklist, and are doing Llama-2 replications (see above).
>
> > data came from the extremely small, cherry-picked set of concepts used in the original Tenenbaum dataset (with N=8)... larger and more systematic datasets are now standard, e.g. Bigalow & Piantadosi
>
> Before performing our study we closely examined the Bigalow & Piantadosi dataset and discussed it with one of the authors of that dataset. We concluded it was too noisy, and that many Mechanical Turk workers were not correctly following the instructions. We suggest examining their data for "powers of 2", which shows that participants failed to even label the provided numbers (16, 32, 2, 8) as being 100% in the concept.
>
> Although small, the Tenenbaum data exhibits important phenomena such as few-shot learning of both rule-based and similarity-based generalizations. It is also canonical and pedagogical, serving as a main example of Bayesian concept learning in the textbook "Machine Learning: A Probabilistic Perspective" (Murphy 2012). Although a bigger dataset would be desirable, we believe the Tenenbaum data effectively shows the basics of the model, before moving onto the bigger logical concepts dataset.
>
> > scaling... outside of toy domains
>
> From the perspective of cognitive modeling, the logical concept data is quite challenging. To the best of our knowledge, there is no other study of logical concept learning in humans which is nearly as broad, high-quality, and high-resolution, as Piantadosi et al. 2016.
>
> > how train-test splits were handled
>
> For Number Game we split the human judgments in 10-way cross-validation, *mixing across concepts.* Originally we thought that with only 8 concepts, fitting a prior while holding out whole concepts would not work. **Based on your question we reran by testing only on held-out concepts, finding that the results change very little: $R^2$ drops from 0.95 to 0.91.** Therefore, the model learns a prior that works for novel concepts / training data. For logical concepts, we followed Piantadosi 2016, which held out specific learning curves run on independent participants. That split shows generalization to new training data, like the new Number Game result given above. Our replication experiments get their test data from running new participants on novel out-of-distribution concepts (training on Piantadosi 2016), again showing that the learned prior can transfer to never-before-seen concepts.
>
> > I would have liked to see a more reasonable neurosymbolic BPL baseline
>
> We appreciate your suggestion of a modern neurosymbolic baseline, and **have now ran a DreamCoder comparison, which gives a decent (but not great) fit to Number Game concepts (attached pdf).**
>
> > 'no proposal dist' model might potentially be competitive if it was given more than 100 samples
>
> We performed a new experiment, finding that **with an order of magnitude more samples ($10^3$), 'no proposal dist' agrees with the human data only at $R^2=.41$, ie, it levels off in fit, although it should eventually trend upward with enough samples.** As the number of samples tends toward infinity, 'no proposal dist' should be just as good as the full model.
>
> >  there is no Bayesian inference at all in this pipeline, and certainly no inference over natural language
>
> We've raised this issue in the global response, and can certainly alter our word choice.
>
> > how the Latent Language model actually worked... did it just take the single maximum likelihood concept from the proposal distribution instead of marginalizing proportional to each sample?
>
> Exactly, it works as you described.

---

> > ### Comment · Reviewer_y5Ba · 2023-08-16
> > **Thanks!**
> >
> > I very much appreciate the careful and thoughtful response, particularly the new results with Llama-2 pointing out the importance of having a strong pre-trained model for the proposal distribution. I think the paper will be greatly strengthened by these changes. I still think this is a good paper (thus the positive score) but given the opportunity for some back-and-forth, I wanted to clarify two points.
> >
> > 1. About the 'brittleness' of the pipeline: I agree that the 'classes' of ablations appropriately map onto the 'joints' of the approach. I was instead trying to note brittleness in the linking function between the idealized mathematical model worked out in section (3) and the many specific *instantiations* or *choices* used to realize the model (e.g. `CodeGen` for the pre-trained prior, `all-MiniLM-L6` for the tuned prior, ` code-davinci-002` as the proposal distribution translating to Python, the tuneable Platt transform as the linking function to Likert ratings, etc.) Each of these choices represent an important 'ancillary assumption' that constrains the interpretation of the results --- the exact same theoretical model with different choices plugged in could do substantially better or worse in practice. Imagine that someone includes this model as a baseline in a future paper; they plug in a set of 'reasonable' off-the-shelf models as the linking functions, and find that it performs very poorly relative to their approach. Would they then be licensed to reject the whole model? Or would you respond "of course it doesn't work if you plug in those, you should have used these." But there are enough 'experimenter degrees of freedom' for each choice that it's not clear what can ultimately be attributed to the core theoretical approach (vis-a-vis section 3) and what is an artifact of the particular bundle of ancillary assumptions used as linking functions. It's ok if it's intended to be an existence proof that some bundle of ancillary assumptions suffices to achieve a certain level of performance, but it's hard to generalize any core principles.
> >
> > 2. About the terminology: I'm sorry to be grumpy about this, but I have to insist that 'inference' is being used in a deeply misleading way here. There is an active community at the intersection of computational cognitive science and ML specifically working on the problem of 'performing Bayesian inference over natural language' (i.e. inferring a posterior `P(concept | natural language) \propto P(natural language | concept) P(concept)` using, e.g., principles of pragmatics and social cognition in the likelihood function). The title and abstract of the paper strongly suggest that the problem has been solved and we are now able to do Bayesian inference over natural language. However, this is not at all the problem addressed by the paper. I believe a large segment of the target audience for this paper (i.e. computational cognitive scientists working on human few-shot concept learning) will be confused or misled by the non-standard evocation of an 'inference' problem here. I'm not familiar with any adjacent literature that uses 'inference' to describe the problem of marginalizing over a pre-existing posterior distribution. I'd be much happier to recommend this paper given a less contentious (but still very cool!) tweak of the title/abstract like "Modeling Human Few-Shot Learning using Amortized Language Models" or "Modeling Human Few-Shot Learning by Translating Linguistic Proposals" or just "Modeling Human Few-Shot Learning using Natural Language" or something.

---

> > > ### Author Response · Authors · 2023-08-19
> > > **Thanks for the engagement! We're revising as follows**
> > >
> > > Thank you for raising interesting points and helping us refine the paper. We're revising as follows:
> > > > the exact same theoretical model with different choices plugged in could do substantially better or worse in practice.
> > >
> > > This is an important point, and relevant to many structured Bayesian cognitive models: there are typically multiple reasonable choices for the prior, likelihood, inference method, and hypothesis space. In our past experience with Bayesian models, good agreement with humans requires at least *some* “tinkering” of these components, and can require sampling budgets larger than what we think humans plausibly process.
> > >
> > > Relative to other Bayesian Program Learners we’ve worked with, this new model required less "tinkering", and vastly smaller sampling budgets. For example, all LLMs reported are the first ones we tried (except for logical concepts: we tried codex before GPT4). We needed less “tinkering” as we largely remove a critical degree of freedom: The design of the structured symbolic hypothesis space itself. We also introduce new degrees of freedom, like the choice of LLM, and new continuous parameters for estimating the prior.
> > >
> > > Some related models, like Rational Rules, have almost no continuous parameters, but require designing a custom discrete hypothesis space. Piantadosi et al. ‘16 shows such model performance depends on that design. We have extra learnable parameters, but remove combinatorial degrees of freedom as a result.
> > >
> > > We'll add this to the discussion:
> > >
> > > **Generalizability of the theoretical framework.** The basics of the model make few commitments, yet instantiating it requires selecting specific language models, engineering prompts and likelihoods, etc. More broadly, a high-resolution cognitive model, particularly a structured Bayesian one, requires domain-specific modeling choices. How much credit should we assign to the general theoretical framing, as opposed to particular engineering decisions? Although our paradigm introduces new degrees of freedom (which LLMs/prompts to use), it removes others (the grammatical structure of the symbolic hypothesis space). On balance, we are cautiously optimistic that the framework will generalize with significantly less domain-specific tinkering, at least for abstract symbolic domains. This optimism is because the framework replaces hand-designed structured hypothesis spaces with pretrained neural models, and because reasonable “default” neural networks worked well across our experiments.
> > > > someone includes this model as a baseline… plug in a set of 'reasonable' off-the-shelf models… find that it performs very poorly relative to their approach. Would they then be licensed to reject the whole model?
> > >
> > > Following the above discussion, we’d feel comfortable with other researchers using the theoretical framework as a baseline and rejecting it if it flunks their data. (And we couldn’t say the same for DreamCoder, BPL, SOAR, etc.) One nuance, though, is that the proposal distribution needs a strong LLM, which we now have hard data to support, thanks to your earlier input.
> > > > the terminology... 'inference' is being used in a deeply misleading way here... [suggesting] the problem of 'performing Bayesian inference over natural language'... using, e.g., principles of pragmatics and social cognition
> > >
> > > Thank you for explaining. We’ll change the title to not include the phrase “Bayesian Inference over Natural Language”, and clarify that our work has nothing to do with the Rational Speech Act model, recursive/social reasoning, etc. Thanks for suggesting possible titles.
> > >
> > > We also use “utterance” in the paper. Do you think that could mislead readers into thinking this is about external communicative language? If so, we’ll globally remove “utterance”.
> > > > I'm not familiar with any adjacent literature that uses 'inference' to describe the problem of marginalizing over a pre-existing posterior distribution
> > >
> > > Respectfully, we’d like to explain why we think the phrase “Bayesian Inference” is consistent with literature on Bayesian Program Learning and Bayesian computational cognitive science.
> > >
> > > We view $q$ as a data-driven proposal distribution. Given that, Lake et al. ‘15 uses “inference” to refer to generating data-driven proposals, which are then weighed by prior and likelihood, like our model. This approximate posterior—the result of inference—is then marginalized over to form predictions, like our model. DreamCoder [Ellis et al. ‘21] also uses this terminology, as does Latent Language [Andreas et al. ‘18] (“inference”, minus the term Bayesian).
> > > Other NeurIPS papers in cognitive modeling have long used similar terminology. From Shi&Griffiths ‘09: “importance sampling provides a simple and efficient way to perform Bayesian inference, approximating the posterior distribution with samples from the prior weighted by the likelihood.”
> > >
> > > Thinking of the raw LLM as a pre-existing amortized posterior is an interesting perspective, but not the only appropriate vocabulary.

---

> > > > ### Comment · Reviewer_y5Ba · 2023-08-20
> > > > **Thanks for the detailed response.**
> > > >
> > > > I appreciate the careful point-by-point responses. While I have to disagree with this characterization of the Bayesian computational cognitive science literature (full disclosure: I am an author on one of the referenced papers), I understand that not all disagreements can be resolved through this channel, and my remaining quibbles are minor relative to the strengths of the paper. I have bumped up my score to reflect the improvements made through the response process.

---

### Author Rebuttal · Authors · 2023-08-09

Thank you all for your reviews, and especially for the encouragement and constructive criticism. Below we summarize some important strengths and weaknesses identified across the reviews, and overview the new results in the attached PDF that address those weaknesses.

**Strengths:**

qc38 describes the work as an “important advance in computational cognitive science” which "identifies a major open question (tractability vs expressivity) and addresses it via a key insight (using natural language is now much more feasible with recent LLM advances)". The other reviewers find the work “promising and potentially influential” [y5Ba], "useful for future researchers" across Bayesian AI and deep learning "as a way to take insights from one school of thought and use them to overcome weaknesses of the other school of thought" [ce9a], and contributing a "super cool result" [8GEn]. Experiments “illuminate both the power and limitations” in light of ”strong baselines” [qc38], yielding "results [that] compare favorably with pother recent approaches that have been well-received at NeurIPS" [y5Ba] and “provides a strong fit to human data” [ce9a]. Reviewers mentioned the ability to explain patterns of human mistakes via "explainable failure modes"; the ability to "generalize to totally novel concepts" [qc38]; and last, the "transfer [of] human priors into a neural model" [ce9a]. From a technical perspective, reviewers describe the computational model as one which "takes classic ideas and “'remixes' and integrates them in an interesting way” [y5Ba] via "multiple non-obvious steps” [qc38].

**Weaknesses, and responses (see also attached PDF):**

- qc38 asks "Is there a stronger or more recent baseline for Number Game, vs Latent Language from 2018?", echoed by y5Ba, who requests "a more reasonable neurosymbolic BPL baseline". We have now run DreamCoder, a 2001 neurosymbolic Bayesian Program Learner. We find a nontrivial gap between DreamCoder and our model (see attached PDF), even when DreamCoder is granted 2 orders of magnitude more test-time samples. This finding further supports the conclusion that, relative to prior BPL frameworks, our model can produce more human-like predictions with far fewer samples.


- y5Ba points out that we use closed-source LLMs, which is a reproducibility hazard. qc38 has a slightly different take, noting "reproducibility issues with using GPT4 seem addressed by including the responses in the software/data release" (indeed: we are including it in the data release, as noted in the author checklist). Further addressing this reproducibility hazard is being done by replicating the results using an open source model, Llama-2. The provided PDF shows results on a Llama-2 Number Game model, showing that open LLMs can be used to build a decent model, but that they are currently slightly worse than OpenAI's LLMs as a proposal distribution. (Llama-2 on logical concepts would take more than 2 weeks to complete on the hardware available to us.)


- y5Ba asks "how to do credit assignment to the many potentially brittle component parts of the pipeline" and requests a broader set of LLMs be tried, including weaker ones for different pipeline stages. As a reminder, the pipeline includes prior, likelihood, and proposal distributions. We ablated the prior by not tuning it, and by disabling it completely (for latent language), and also ablated the proposal distribution, but we never ablated the likelihood. The attached PDF shows a new likelihood ablation, revealing that the likelihood is very important. To understand what happens when the individual components are merely made weaker instead of ablated entirely, the new Llama-2 results include data for when individual components are replaced with the weaker Llama-2. The new data show that weaker LLMs can implement effective likelihood models, but stronger LLMs are important for proposal distributions, especially in the low-sample regime. Although the Llama-2 results do not change the scientific conclusions, they are helpful in understanding how to practically engineer systems like ours using off-the-shelf tools.


**Potentially Revised Terminology:** Reviewer y5Ba suggests that the framing as "Bayesian inference over natural language" is technically incorrect, preferring to think of pretraining the LLM as "inference", and our model as predicting based on that already inferred distribution. We're happy to revise our terminology, and understand that Bayesian vocabulary can be quite nuanced (and occasionally contentious!). Right now, we think our original terminology is consistent with adjacent literature, but we can go with whatever terminology the reviewers collectively feel is best.

---

### Decision · Program_Chairs · 2023-09-21

**Decision:**

Accept (oral)

**Comment:**

This paper presents a method that synthesizes computational cognitive science modeling with large language models, crossing discipline boundaries and making a creative new contribution. Reviewers all agree it should be published, with some very strongly advocating for its value. Authors were responsive during the rebuttal, addressing  the reviewer suggestions. Should definitely be presented at NeurIPS, likely as a spotlight or oral.